# Early-life-trauma triggers interferon-β resistance and neurodegeneration in a multiple sclerosis model via downregulated β1-adrenergic signaling

Yee Ming Khaw [ORCID] [1,2], Danish Majid[1,3], Sungjong Oh[1,3], Eunjoo Kang[1,2] & Makoto Inoue [ORCID] [1,2 ✉]

Environmental triggers have important functions in multiple sclerosis (MS) susceptibility, phenotype, and trajectory. Exposure to early life trauma (ELT) has been associated with higher relapse rates in MS patients; however, the underlying mechanisms are not well-defined. Here we show ELT induces mechanistic and phenotypical alterations during experimental autoimmune encephalitis (EAE). ELT sustains downregulation of immune cell adrenergic receptors, which can be attributed to chronic norepinephrine circulation. ELT-subjected mice exhibit interferon-β resistance and neurodegeneration driven by lymphotoxin and CXCR2 involvement. These phenotypic changes are observed in control EAE mice treated with β1 adrenergic receptor antagonist. Conversely, β1 adrenergic receptor agonist treatment to ELT mice abrogates phenotype changes via restoration of immune cell β1 adrenergic receptor function. Our results indicate that ELT alters EAE phenotype via downregulation of β1 adrenergic signaling in immune cells. These results have implications for the effect of environmental factors in provoking disease heterogeneity and might enable prediction of long-term outcomes in MS.

---

[1] University of Illinois at Urbana-Champaign Department of Comparative Biosciences, 2001 South Lincoln Avenue, Urbana, IL 61802, USA. [2] University of Illinois at Urbana-Champaign Neuroscience Program, 405 North Matthews Avenue, Urbana, IL 61801, USA. [3] University of Illinois at Urbana-Champaign School of Molecular and Cell Biology, 407 South Goodwin Avenue, Urbana, IL 61801, USA. ✉email: makotoi@illinois.edu

Multiple sclerosis (MS) is characterized by demyelination of neurons in the central nervous system (CNS), and an estimated 2.5 million individuals currently have MS worldwide[1]. MS is thought to be a disease of diverse subtypes characterized by distinct clinical presentations, immune signatures, and reactivity and/or resistance to pharmaceutical treatments[2,3]. Patients with MS experience various neurological symptoms that differ in type and severity[3,4]. While its etiology remains debated, MS is recognized as a multifactorial heterogeneous disease with both genetic and environmental causes. To illustrate, environmental factors like Epstein-Barr virus[5], vitamin D exposure levels[6], cigarette smoking[7,8], obesity[9], and stress[10] are thought to converge with genetic determinants to influence the manifestation of MS progression.

Childhood trauma or early-life trauma (ELT), characterized by events such as emotional neglect or physical abuse, increases MS susceptibility later in life along with MS relapse rates and likelihood of MS-related hospitalization[11,12]. Animal studies using multiple model of MS in rats and mice also indicate an association between ELT and disease severity[13–16]. ELT is proposed to alter immune cell properties in adults and is associated with increased risk of chronic illnesses[17–19]. However, the mechanism through which ELT alters MS disease progression remains elusive. Importantly, existing studies did not investigate changes in disease phenotype characterized by immune cell properties that influence drug efficacy and disease symptoms.

Here we use C57BL/6J mice and the experimental autoimmune encephalitis (EAE) model to demonstrate that ELT induces not only increases in severity and susceptibility in EAE but also induces a phenotype shift characterized by interferon-β insensitivity, neuronal damage in the spinal cord, in addition to membrane-bound lymphotoxin and CXCR2 involvement. We also observe downregulated β1 adrenergic receptor (AR) in immune cells of ELT-EAE mice. Rescue of β1 receptor function restores interferon-β sensitivity and protects against CNS neuronal damage. Therefore, our results indicate that phenotype changes in mice subjected to ELT can be explained by downregulated adrenergic signaling in immune cells.

## Results

### ELT affects EAE susceptibility and severity in male and female mice.
ELT or stress was modeled by neonatal maternal separation and sterile phosphate-buffered saline (PBS) injections to induce emotional and physical distress (see "Methods"). EAE scores of ELT mice were compared against mice that did not receive any handling (control mice) (Fig. 1a). ELT-EAE male mice showed slightly early induction of EAE disease with prolonged severe paralysis in comparison to control EAE mice (Fig. 1b). Quantification of cumulative disease severity from 0- to 30-day post-EAE induction (dpi) revealed that male ELT-EAE mice had significantly exacerbated disease in comparison to EAE mice (Fig. 1b). Female ELT mice also developed EAE at an earlier time and prolonged severe disease, when compared with non-ELT control mice (Fig. 1c). To examine if the condition of ELT (combination of neonatal maternal separation and sterile saline injection) is necessary to provoke severe disease, we compared disease severity of control EAE mice against EAE mice subjected to maternal separation or sterile PBS injection alone. In both male and female mice, maternal separation alone, but not PBS injection, resulted in increased EAE severity during disease chronic and late phase when compared to control EAE mice (Supplementary Fig. 1a, b). However, EAE mice subjected to maternal separation alone did not show significant early induction, compared with control mice (Supplementary Fig. 1a, b). Because EAE is a demyelinating disease of the CNS, Luxol-fast blue myelin

staining was conducted using lumbar spinal cord samples to reveal that ELT-EAE mice had significantly more severe demyelination in the spinal cord compared to control EAE mice at 20 dpi (Fig. 1d). We also compared demyelination status between control EAE mice and EAE mice subjected to maternal separation or PBS injection only. No significant difference in demyelination was observed between groups (Supplementary Fig. 1c). These data suggest that the entire ELT-EAE phenotype is exclusively presented by the specific conditions of subjecting B6 mice to neonatal maternal separation and sterile PBS injection followed by EAE induction in adulthood.

To test the hypothesis that ELT increases EAE susceptibility, we titrated the amounts of heat-killed Mycobacterium tuberculosis (Mtb) (adjuvant) used in complete Freund's adjuvant to identify a weak disease induction dose in control mice (Supplementary Fig. 1d). While only 17% of control mice displayed an EAE score ≥1 over the course of 30 days using a low dose (50 μg/mouse) of Mtb for EAE induction, all ELT mice developed scores ≥1 (Fig. 1e). Taken together, these results suggest that ELT increases EAE severity with no sex specificity and increases mice susceptibility to EAE with low-dose Mtb.

### ELT alters peripheral and CNS immune cell profiling.
ELT affects immune system development and alter an organism's response to future immune challenges[20,21]. In young adult mice without EAE induction, there was no difference of immune cell population observed in lymph nodes of control and ELT mice (Fig. 2a). However, upon EAE induction, ELT mice had significantly increased numbers of CD4+ T cells, CD8+ T cells, and macrophage in axillary and inguinal lymph nodes at 10 dpi compared to non-ELT controls (Fig. 2b and Supplementary Fig. 2a). Similar population similarities and differences before and after EAE induction between control EAE and ELT-EAE mice, respectively, were found in the spleen (Supplementary Fig. 2b). Notably, in mice subjected only to either maternal separation or sterile saline injection during the neonatal phase and induced with EAE in adulthood, there was no significant change in lymph node immune cell numbers (Supplementary Fig. 2c). Again, these data suggest that the entire ELT-EAE phenotype is a manifest of the specific conditions of subjecting B6 mice to neonatal maternal separation and sterile PBS injection followed by EAE induction in adulthood.

To understand the mechanism of T cell upregulation, we first examined co-stimulatory molecule expression on antigen-presenting cells (APCs) and secondly evaluated T cell proliferative abilities. 4-1BBL and CD80, but not CD86, B7-H2, or MHC class II, were significantly increased on DCs from lymph nodes of ELT-EAE mice at 10 dpi compared to controls (Fig. 2c and Supplementary Fig. 2d, e). Macrophages isolated from ELT-EAE mice also had significant upregulation of 4-1BBL and CD80 (Supplementary Fig. 2f). In mice subjected only to either maternal separation or sterile saline injection during the neonatal phase and induced with EAE in adulthood, there was no significant change in APC expression of co-stimulatory molecules CD80 and 4-1BBL (Supplementary Fig. 2g). To evaluate whether immune cells derived from ELT-EAE mice were more proliferative as the evidence of increased APC co-stimulatory molecule would suggest, we examined CD4+ T cell proliferation in vitro. CD4+ T (CD3+CD4+) cells were isolated from the lymph nodes of control EAE and ELT-EAE mice, labeled with CFSE, and cultured with MOG tetramer to allow observation of antigen-specific T cell proliferation. After 72 h, CD4+ T cells derived from ELT-EAE mice exhibited more proliferation as compared to control EAE condition as detected by fluorescent dye dilution (Supplementary Fig. 2h). To determine if altered properties of ELT-EAE APCs

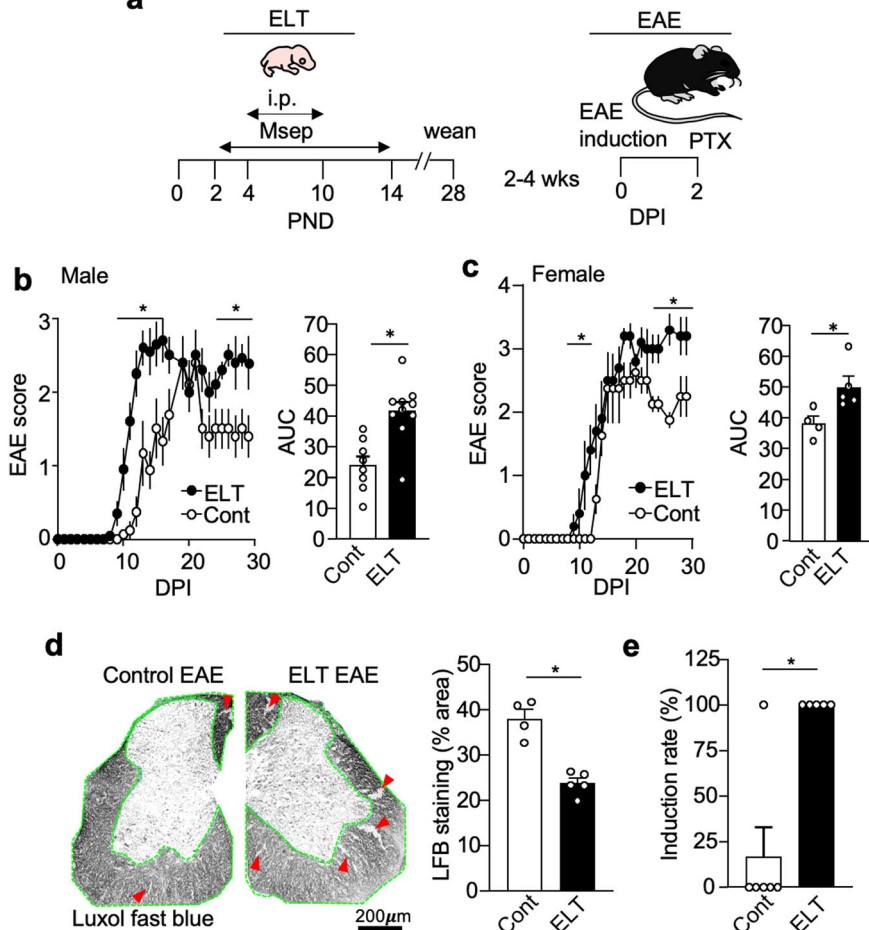

**Fig. 1 ELT impacts EAE susceptibility, severity, and duration in both male and female mice. a** Schematic representation of experimental procedure. PND and DPI indicate postnatal day and day post-EAE induction, respectively. **b** Mean clinical EAE scores and cumulative scores of male mice (Cont EAE: $n = 8$, ELT-EAE: $n = 10$). **c** Mean clinical EAE scores and cumulative scores of female mice (Cont EAE: $n = 4$, ELT-EAE: $n = 5$). **d** Spinal cord sections of EAE mice stained for myelin using Luxol-fast blue at 20 dpi (Cont EAE: $n = 4$, ELT-EAE: $n = 5$) with quantitative analysis of stained area. Green outline indicates analyzed white matter region of interest. Red arrows indicate representative demyelination regions. **e** EAE induction rate among mice treated with low-dose Mtb (50 μg/mouse, Cont EAE: $n = 6$, ELT-EAE: $n = 5$). Results are representatives of at least two independent experiments. Each dot represents averaged data per animal. Data are represented as mean ± SEM. Two-tailed Student's t-test, *$P < 0.05$. Exact $P$ values for asterisks: **b** 0.0010, **c** 0.0288, **d** 0.0004, and **e** 0.0112.

contributed to increased T cell numbers in draining lymph nodes, we co-cultured CD4+ T cells from naïve TCR2D2 mice (MOG-specific T cell transgenic mice) with DCs harvested from ELT or control mice with EAE at 3 dpi. T cell proliferation was significantly upregulated in ELT-EAE mice when naïve T cells were co-cultured with ELT-EAE-derived DCs compared to naïve T cells that were co-cultured with control EAE-derived DCs (Supplementary Fig. 2i). Additionally, in vitro co-culture experiments showed that ELT-EAE-derived CD4+ T cells produce significantly a greater abundant of cytokines, namely, IL-17, IFNγ, and GM-CSF, but not IL-2, in the presence of ELT-EAE-derived DCs upon incubation with MOG35-55 peptide compared to control EAE-derived T cells that were co-culture with control EAE-derived DCs (Fig. 2d and Supplementary Fig. 2j). Under the absence of DC, T cells yielded undetected amounts of cytokine from culture supernatant (Fig. 2d and Supplementary Fig. 2j). These results suggest that the heightened T cell immune response to EAE generated in ELT mice can be attributed amplified antigen-dependent interactions between innate and adaptive immune cells.

Neuroinflammation characterized by glia activation and peripheral immune cell infiltration to the CNS is important for

EAE and MS pathology[22,23]. ELT-EAE mice had significantly more spinal cord-infiltrated CD4+ T cells at 10 dpi (disease onset), 18 dpi (disease peak), and 25 dpi (disease chronic phase) when compared to control EAE mice (Fig. 2e and Supplementary Fig. 2k). T cell subtypes were then characterized based on cytokine-producing profiles. Significant upregulation of IL-17- and/or IFNγ-producing CD4+ T cells was observed in the spinal cord of ELT-EAE mice at 18 and 25 dpi (Fig. 2f and Supplementary Fig. 2l, m). Additionally, microglia population was significantly greater in the spinal cord of ELT-EAE mice at both time points 18 and 25 dpi when compared with control EAE mice (Fig. 2g and Supplementary Fig. 2k, n). Notably, significant increases in spinal cord-infiltrated myeloid cells such as neutrophil and macrophages were found at 18 dpi but not at 25 dpi (Supplementary Fig. 2n). These findings demonstrate that the increased severity and early induction of disease in ELT-EAE mice may be attributed to heightened immune cells in the periphery and CNS which likely results from immune cell property changes observed in peripheral lymphoid organs.

**Severe neuron damage in ELT-EAE mice.** To test the impact of ELT on EAE-induced neuronal damage, we examined morphological

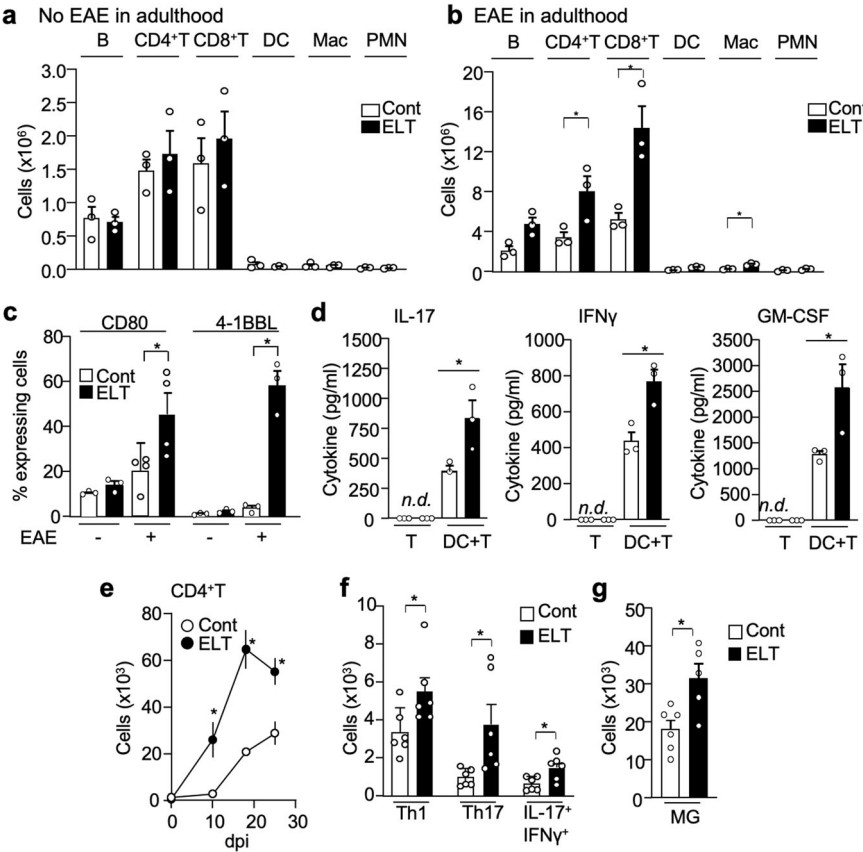

**Fig. 2 ELT alters peripheral and CNS immune cell profiling. a, b** Absolute immune cell numbers (B cell: CD19+, CD4+T cell: CD3+CD4+, CD8+T: CD3+CD8+, dendritic cell: CD11b+CD11c+, macrophage: CD11b+CD11c−, neutrophil: CD11b+Ly6Ghigh) in control and ELT mice without (**a**) and with (**b**) EAE. Cells were isolated from draining lymph nodes were isolated from EAE mice at 10 dpi ($n = 3$ per group). **c** Percentage of co-stimulatory molecule (CD80 and 4-1BBL) expressing dendritic cells (CD80: $n = 4$ per group: 4-1BBL: $n = 3$ per group). **d** Amount of cytokines IL-17, interferon γ (IFNγ), and GM-CSF in supernatant derived from culture conditions 72 h post culture initiation. CD4+ T cells were isolated from draining lymph nodes of ELT-EAE or control EAE mice on 10 dpi to be supplemented with MOG (10 μg/ml) and cultured alone or with isolated dendritic cell isolated from ELT-EAE or control EAE mice ($n = 3$ per group). White and black bars represent data from control and ELT conditions, respectively. **e** Absolute numbers of CD4+T (CD45+CD3+CD4+) cells in spinal cord isolated from EAE mice on 0, 10, 18, and 25 dpi ($n = 5$ per group). **f** Absolute numbers of IL-17+-secreting CD4+T (Th17), interferon γ+-secreting CD4+T cells (Th1), and IL-17+ interferon γ+ CD4+T cells in the spinal cord of EAE mice at 25 dpi ($n = 6$ per group). **g** Absolute numbers of microglia (MG: CD45lowCD11b+) in the spinal cord at 25 dpi ($n = 5$ per group). Each dot represents averaged data per animal. Data are represented as mean ± SEM. Two-tailed Student's t-test, *$P < 0.05$. Exact P values for asterisks (from left to right): **b** 0.0497, 0.0172, 0.0050; **c** 0.0456, 0.0013; **d** 0.0478, 0.0164, 0.0470; **e** 0.005, 0.0004, 0.0062; **f** 0.045, 0.022, 0.021; and **g** 0.0140.

abnormalities in motor neurons using Golgi–Cox staining. We compared the percentage of area stained by Golgi–Cox in the ventral horn of lumbar 3–6 spinal cord slices at 30 dpi (Fig. 3a). While Golgi–Cox staining did not differ between control and ELT mice before EAE induction, staining was significantly reduced in ELT-EAE mice (Fig. 3b). On the other hand, percentage area stained by Golgi–Cox in the ventral horn of lumbar spinal cord was not significantly reduced in mice subjected to maternal separation or PBS injection only (Supplementary Fig. 3a). We then sought to determine dendritic spine density using the recently reported Golgi-visualizing confocal reflection super-resolution method (CRSR)[24]. Reduced spine density was found in the dendrites of neurons that reside in the lumbar spinal cord ventral horn in control EAE mice and ELT-EAE mice at 30 dpi, compared to naïve and ELT mice without EAE (Fig. 3c, d and Supplementary Fig. 3b). Notably, spine density loss in ELT-EAE mice was significantly worse than in control EAE mice (Fig. 3c, d and Supplementary Fig. 3c). To further evaluate the functional status of spinal cord neurons, we conducted FluoroJade-C staining to detect degenerating neurons[25]. Neurodegenerative signals were significantly more prominent in the spinal cord lumbar region

of ELT-EAE mice compared to control EAE mice (Fig. 3e). Further, we quantitated motor neurons in lumbar spinal cord using immunohistochemistry to label choline acetyltransferase (ChAT). Number of ChAT-positive motor neurons was significantly reduced in ELT-EAE mice, compared with controls (naïve mice, naïve ELT mice, and control EAE mice) (Fig. 3f and Supplementary Fig. 3c). Next, we assessed activation status of microglia in the lumbar spinal cord region by examining soma area (Fig. 3g). A significant increase in Iba1+ cells of large soma areas, indicative of microglial activation, was observed in ELT-EAE mice when compared with control EAE mice (Fig. 3h). Collectively, these results demonstrate that ELT alters CNS neuron and microglia properties upon EAE induction.

**Altered drug sensitivity and distinct immune markers in ELT-EAE mice.** Interferon-β sensitivity is a biomarker for EAE disease alterations, with some mirroring clinical evidence[26,27]. While interferon-β suppressed EAE development in control mice, the same treatment did not suppress EAE development in male and female mice subjected to ELT (Fig. 4a, b). Mice subjected only to maternal separation and subjected to EAE in adulthood were also

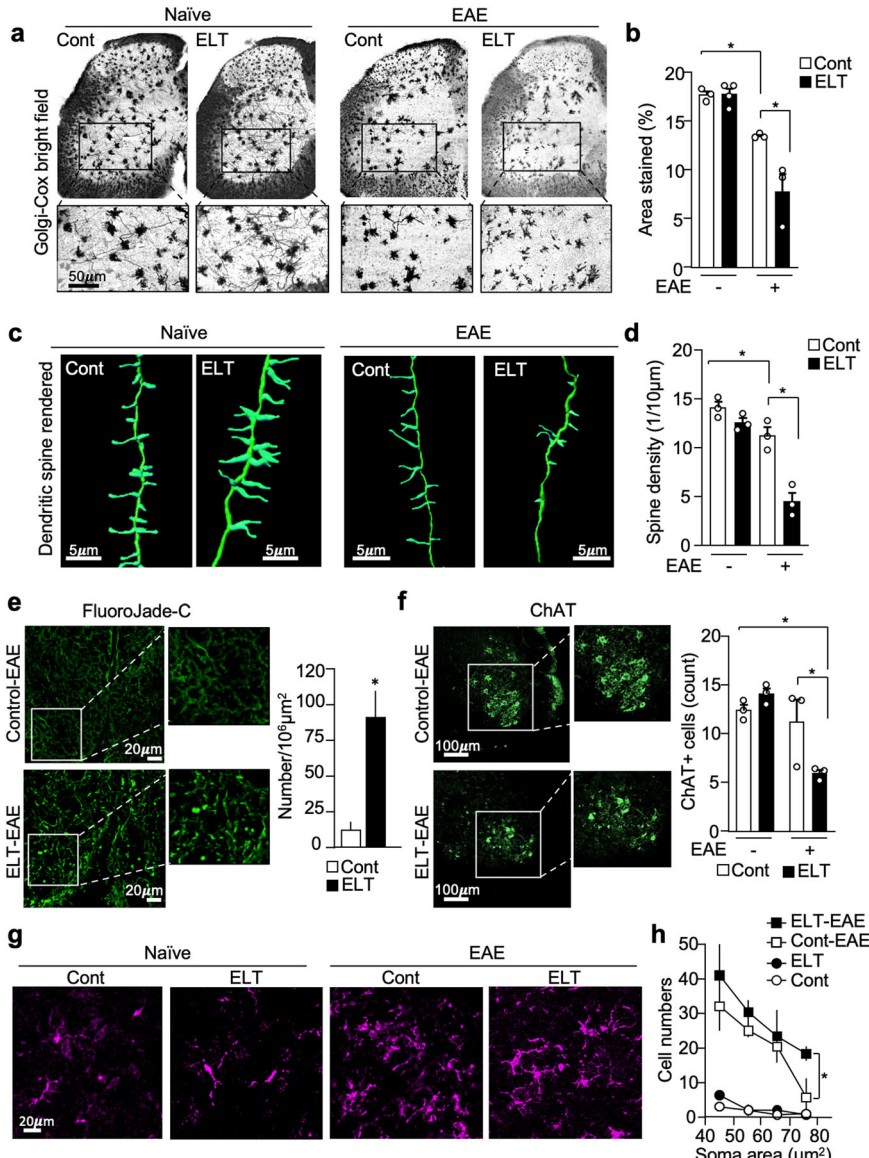

**Fig. 3 Severe neuron damage in ELT-EAE mice. a**, **b** Golgi–Cox stained lumbar spinal cord sections of EAE mice isolated at 30 dpi (**a**) with quantification of area stained/per field (**b**) (Cont naïve: $n = 3$, ELT naïve: $n = 4$, Cont EAE: $n = 3$, ELT-EAE: $n = 3$). **c**, **d** CRSR imaging of Golgi–Cox-stained dendrites in lumbar spinal cord ventral horn region (**c**) with quantification of dendritic spine density at 30 dpi (**d**) ($n = 3$ animals/group, 60 dendrites/group). **e** FluoroJade-C-stained lumbar spinal cord sections of EAE mice isolated at 20 dpi with quantification of area stained/per field (Cont EAE: $n = 7$, ELT-EAE: $n = 4$). **f** ChAT+ neuron-stained lumbar spinal cord sections of EAE mice isolated at 30 dpi ($n = 3$ per group). **g** Representative image of Iba1+ cells in spinal cord ventral horn at 30 dpi. Representative image was randomly selected from pooled images from two independent staining experiments. **h** Iba1+ cell count categorized by soma size in the lumbar spinal cord ventral horn (L5–6) at 30 dpi ($n = 3$ per group). Each dot represents averaged data per animal. Data are represented as mean ± SEM. Two-tailed Student's $t$-test, $*P < 0.05$. Exact $P$ values for asterisks (from left to right): **b** 0.0005, 0.0350; **d** 0.0382, 0.0102; **e** 0.0007; **f** 0.0007, 0.0072; **h** 0.0204.

resistant to interferon-β whereas mice subjected to neonatal sterile PBS injection only were responsive to interferon-β treatment (Supplementary Fig. 4). Inoue et al. reported that while interferon-β-sensitive EAE development is dependent on NLRP3-inflammasome-mediated IL-1β[28,29], interferon-β-resistant EAE is driven by membrane-bound lymphotoxin (mLT) and lymphotoxin β receptor (LTβR) function, not inflammasome-mediated IL-1β[27]. To examine if the interferon-β-resistant profile of ELT-EAE disease is mediated by an NLRP3-inflammasome-independent pathway, we monitored EAE symptoms in $Asc^{-/-}$ mice with and without ELT. ASC is an adaptor molecule that plays a pivotal role in inflammasome assembly mainly implicated in innate immune cells. ASC depletion did not impact the disease

development of ELT-EAE mice whereas ASC is necessary in control EAE disease progression (Fig. 5a), suggesting that the development of ELT-EAE was indeed NLRP3-inflammasome-independent. Additionally, while similar amounts of IL-1β was found in the serum of ELT and control mice without EAE induction, IL-1β was significantly reduced in ELT mice with EAE when compared to control mice with EAE at 10 dpi (Fig. 5b). This suggests the lack of function of IL-1β in promoting ELT-EAE disease development. These results demonstrate that ELT-EAE mice were resistant to interferon-β in an NLRP3-inflammasome-independent pathway.

Next we asked if markers of NLRP3-inflammasome-independent interferon-β resistance share the same critical

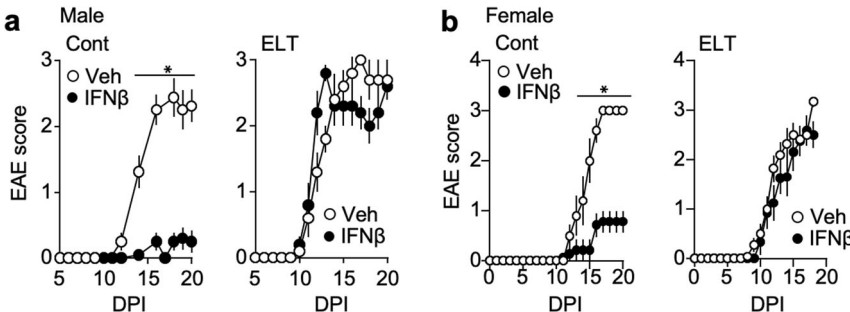

**Fig. 4 ELT-EAE mice are interferon-β resistant. a, b** Mean clinical EAE scores of control EAE and ELT-EAE male (**a**) and female (**b**) mice with or without interferon β (IFNβ) treatment (Male Cont-Vehicle: $n = 8$, Male Cont-IFNβ: $n = 10$, Male ELT-Vehicle: $n = 5$, Male ELT-IFNβ: $n = 5$, Female Cont-Vehicle: $n = 5$, Female Cont-IFNβ: $n = 7$, Female ELT-Vehicle: $n = 5$, Female ELT-IFNβ: $n = 5$). Interferon-β was treated from 0 to 10 dpi every other day. All data represent mean ± SEM. Two-tailed Student's $t$-test, *$P < 0.05$. Exact $P$ values for asterisks are derived from AUC (from left to right): **a** <0.0001, **b** <0.0001.

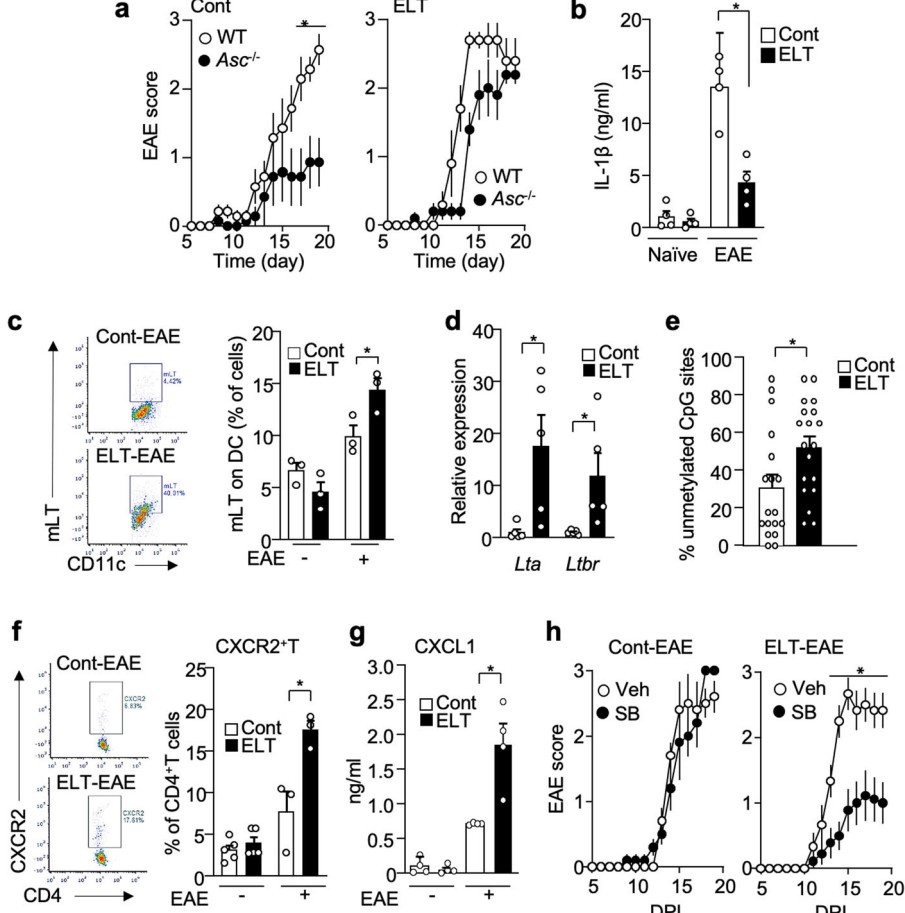

**Fig. 5 Distinct immune markers in ELT-EAE mice. a** EAE scores after EAE induction in control- and ELT mice of wild type and $Asc^{-/-}$ mice (Cont-WT: $n = 4$, Cont-$Asc^{-/-}$: $n = 5$, ELT-WT: $n = 6$, ELT-$Asc^{-/-}$: $n = 5$). **b** Serum levels of IL-1β at 10 dpi (Cont naïve: $n = 4$, ELT naïve: $n = 4$, Cont EAE: $n = 3$, ELT-EAE: $n = 3$). **c** Percentage of mLT-expressing dendritic cells (DC) at 10 dpi ($n = 3$ animals/group). **d** Expression of $Lta$ and $Ltbr$ mRNA in DC obtained from lymph nodes of control EAE and ELT-EAE mice at 10 dpi ($n = 5$ animals/group). **e** Percentage of unmethylated CpG sites in the promoter region of LT gene in dendritic cells isolated from lymph nodes of control and ELT mice without EAE at 10 dpi ($n = 18$ colonies). **f** Percentage of CXCR2-expressing CD4⁺T cells at 10 dpi (Cont naïve: $n = 5$, ELT naïve: $n = 5$, Cont EAE: $n = 3$, ELT-EAE: $n = 3$). **g** Serum levels of CXCL1 at 10 dpi ($n = 4$ animals/group). **h** EAE score of mice treated with CXCR2 inhibitor SB225002 (SB) or saline (Cont-Vehicle: $n = 5$, Cont-SB: $n = 5$, ELT-Vehicle: $n = 6$, ELT-SB: $n = 9$). Male mice were used for experiments. Each dot represents averaged data per animal. Data are represented as mean ± SEM. Two-tailed Student's $t$-test, *$P < 0.05$. Exact $P$ values for asterisks (from left to right): **a** 0.0131; **b** 0.0032; **c** 0.0449; **d** 0.0252, 0.0447; **e** 0.0210; **f** 0.0221; **g** 0.0088; and **h** 0.0084.

cellular markers as observed previously in the animal model and in interferon-β-resistant clinical cohort[27]. Namely, Inoue et al. reported that interferon-β-resistant EAE model was characterized by heightened membrane lymphotoxin (mLT: a heterotrimer containing LTα and LTβ[30]) expression in APCs and upregulated

chemokine receptor CXCR2 expression in T cells. Consistent with experimental evidence, peripheral blood mononuclear cells isolated from interferon-β non-responder patients with MS highly express LTβR and CXCR2 compared to interferon-β responder patients[27]. In comparison to control EAE mice,

ELT-EAE mice higher expression of mLT protein on DCs and macrophages (Fig. 5c and Supplementary Fig. 5a). Transcription of *Lta* and *Ltbr* are also highly upregulated in DCs derived from ELT-EAE mice when compared with control EAE mice (Fig. 5d). Additionally, *Lta* promoter region in DCs isolated from ELT-EAE mice had a significantly increased unmethylated status compared to control EAE mice (Fig. 5e and Supplementary Fig. 5b). Further, we found increased number of CXCR2-expressing CD4+T cells in the lymph nodes of ELT-EAE mice at 10 dpi (Fig. 5f) and higher amounts of CXCL1, a ligand for CXCR2, in serum of ELT-EAE mice compared to control EAE mice (Fig. 5g). DCs and macrophages in lymph nodes of ELT-EAE at 10 dpi also had upregulated expressions of CXCR2 when compared to control condition (Supplementary Fig. 5c). While SB225002, a CXCR2 antagonist, did not impact EAE disease progression in control mice, as previously reported[27], it significantly ameliorated EAE in ELT mice (Fig. 5h). Taken together, these data suggest that altered disease phenotype in ELT-EAE mice is mediated by an alternative inflammatory mechanism that involves mLT and CXCR2 signaling.

**ELT downregulates adrenergic signaling in immune cells**. ELT causes short-term or enduring changes in the systemic sympathetic nervous system (SNS)[31] and hypothalamic–pituitary–adrenal axis (HPA) activity[32] which were shown to have major implications on immune cell property and response to stimuli[33,34]. Levels of nor-epinephrine, an SNS neurotransmitter, in blood plasma of 2- to 6-week-old ELT mice were increased compared to controls (Fig. 6a). However, corticosterone level, which is a marker for HPA activity, were not significantly different between ELT and control mice without EAE (Supplementary Fig. 6a). The pattern of upregulated systemic norepinephrine in ELT mice was also observed at 10 dpi under EAE condition (Supplementary Fig. 6b). Additionally, norepinephrine levels in the inguinal and axillary lymph nodes of 4-week-old ELT mice were significantly heightened compared to controls (Fig. 6b).

Chronic SNS hyperactivity results in downregulation of AR signaling and expression levels in immune cells[35–37]. Therefore, we evaluated AR expression in DCs and CD4+T cells from lymph nodes of ELT mice. Gene expression of all classes of ARs except β3-AR (i.e., α1, α2, β1, and β2) were detected in both DCs and CD4+T cells isolated from lymph nodes of control mice (Supplementary Fig. 6c). Notably, expression was higher in DCs than CD4+T cells (Supplementary Fig. 6c). α1-, α2-, β1-, and β2-AR expression were significantly downregulated on DCs (Fig. 6c and Supplementary Fig. 6d) and CD4+T cells (Fig. 6d and Supplementary Fig. 6e) in the lymph nodes of ELT mice compared to control mice.

To examine if reduced level of AR expression affects AR signaling in immune cells, DC and CD4+T cell isolated from control and ELT mice were treated with β1-AR agonist (Xamoterol) and were examined for the amount of downstream signaling molecule cyclic-AMP (cAMP). While treatment of β1-AR agonist increased cAMP levels in DC and CD4+T cell of control mice, the treatment failed to upregulated cAMP levels in both DC and CD4+T cell of ELT mice (Fig. 6e, f). These results suggest that ELT-mediated sustained SNS activation induces desensitization and downregulation of ARs in DCs and CD4+T cells.

**β1-AR antagonism is sufficient to reproduce ELT phenotypes in EAE**. To determine the role of reduced AR signaling in mediating key phenotypes in ELT-EAE mice, we systemically treated control mice with AR antagonists against α1 (prazosin)[38], α2 (atipamezole)[39], or β1/β2 (propranolol) within the first 10 days of EAE induction (disease induction phase). Injections of α1- and α2-AR antagonists to control EAE mice did not increase EAE severity, when compared with vehicle-treated groups

(Fig. 7a). In contrast, β-AR antagonist treatment to control mice was sufficient to induce severe and prolonged EAE (Fig. 7a). In addition, control EAE mice treated with β-AR antagonist were resistant to interferon-β (Fig. 7b). Specifically, interferon-β-resistant EAE was observed in control EAE mice treated with metoprolol tartrate, a β1-AR antagonist[40], but not with ICI 118,551, a β2-AR antagonist[41] (Fig. 7b). Furthermore, we observed significant neuron damage reflected by reduction of Golgi–Cox staining in the ventral horn region of lumbar spinal cord sections in β-AR antagonist and β1-AR antagonist-treated ELT mice (Fig. 7c), suggesting that the inhibition of β1-AR signaling in disease induction phase was sufficient to produce interferon-β-resistant phenotype with severe neuron damage in the disease chronic phase.

Cell proliferation assay revealed that β1-AR antagonist treatment to EAE mice significantly increased cell proliferation (Supplementary Fig. 7a). Additionally, co-culture supernatant of DC and CD4+ T cells isolated from β1-AR antagonist-treated control mice had significantly increased production of IL-17, IFNγ, and GM-CSF upon incubation with MOG35-55 peptide, compared with the vehicle-treated group (Fig. 7d). Furthermore, β1-AR and β-AR antagonist treatment in control EAE mice was sufficient to induce mLT upregulation on DCs and CXCR2 upregulation on CD4+ T cells (Fig. 7e and Supplementary Fig. 7c), which are mechanistic signatures of interferon-β-resistant EAE phenotype. Further, higher amounts of CXCL1 were detected in EAE mice with β1-AR antagonist treatment when compared to EAE mice that did not receive treatment (Fig. 7f). These results suggest that disease phenotypes of ELT-EAE are a result of dysfunctional β1-AR signaling.

**β1-AR agonist treatment is sufficient to rescue ELT-EAE subtype**. We attempted to rescue β1 adrenergic signaling in ELT-EAE mice by administering β1-AR agonist before EAE disease onset. β1-AR agonist (Xamoterol)[42] treatment was sufficient to revert ELT-mediated interferon-β insensitivity back to an interferon-β-sensitive phenotype (Fig. 8a). Additionally, β1-AR agonist treatment in ELT-EAE mice was sufficient to inhibit mLT upregulation on DCs and CXCR2 upregulation on CD4+ T cells (Fig. 8b). Furthermore, we observed significant rescue of neuron damage reflected by increase of Golgi–Cox staining in the ventral horn region of lumbar spinal cord sections in β1-AR agonist-treated ELT-EAE mice when compared with ELT-EAE (Fig. 8c). Spine density loss was significantly rescued by β1-AR agonist treatment in ELT-EAE mice (Supplementary Fig. 8). To determine if such rescue function of β1-AR agonist in ELT mice is dependent on DC and T cells, DC- and T cell-specific β1-AR knockout mice were subjected to ELT, EAE, and β1-AR agonist treatment. While control β1-AR floxed ELT mice with β1-AR agonist treatment remains sensitive to interferon-β, DC-specific and T cell-specific β1-AR knockout ELT mice canceled the β1-AR agonist-induced rescue function on interferon-β sensitivity (Fig. 8d). Additionally, both DC-specific and T cell-specific β1-AR knockout ELT mice also canceled the β1-AR agonist-induced rescue function on mLT upregulation on DCs and CXCR2 upregulation on CD4+ T cells (Fig. 8e). Interestingly, β1-AR agonist-induced rescue function on neuron damage in the ventral horn region of lumbar spinal cord sections was canceled in T cell-specific β1-AR knockout ELT mice, but not in DC-specific β1-AR knockout ELT mice (Fig. 8f). These results demonstrate that reduction of β1-AR signal in immune cells of ELT-EAE mice induces EAE phenotype shift.

**β1-AR agonist treatment prevents upregulation of *Lta* in BMDC in vitro**. ELT mice showed upregulation of mLT in DCs

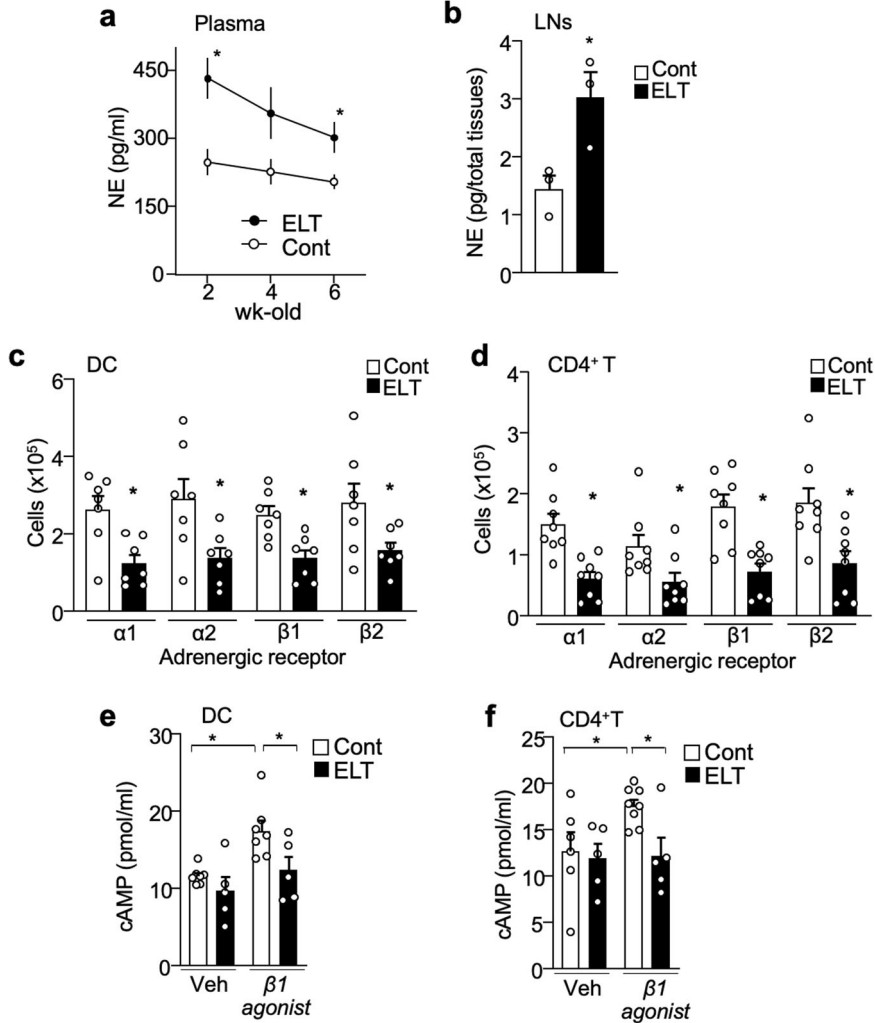

**Fig. 6 ELT downregulates adrenergic signaling in immune cells. a** Plasma levels of norepinephrine (NE) in 2-, 4-, and 6-week-old control and ELT mice (Cont-2wk-old: $n = 5$, Cont-4wk-old: $n = 7$, Cont-6wk-old: $n = 8$, ELT-2wk-old: $n = 4$, ELT-4wk-old: $n = 4$, ELT-6wk-old: $n = 9$). **b** Inguinal and axillary lymph node levels of norepinephrine (NE) in 4-week-old control and ELT mice ($n = 3$ animals/group). **c**, **d** Number of **c** dendritic cells (DC, $n = 7$ animals/ group) and **d** CD4+T cells ($n = 8$ animals/group) expressing adrenergic receptors: α1, α2, β1, and β2 in lymph nodes of 3–4-week-old control and ELT mice. **e**, **f** Amount of cAMP derived from DC and CD4+T cells treated with vehicle or β1-AR agonist (Cont-DC-Veh: $n = 7$, ELT-DC-Veh: $n = 5$, Cont-DC-β1 agonist: $n = 7$, ELT-DC-β1 agonist: $n = 5$, Cont-CD4+T-Veh: $n = 6$, ELT-CD4+T-Veh: $n = 5$, Cont-CD4+T-β1 agonist: $n = 8$, ELT-CD4+T-β1 agonist: $n = 5$). Immune cells were isolated from control and ELT mice without EAE. Male mice were used for experiments. Each dot represents averaged data per animal. Data are represented as mean ± SEM. Student's t-test, *$P < 0.05$. Exact P values for asterisks (from left to right): **a** 0.0093, 0.0239; **b** 0.0353; **c** 0.0068, 0.0245, 0.0041, 0.0446; **d** 0.0009, 0.0335, 0.0008, 0.0080; **e** 0.0017, 0.0435; and **f** 0.0293, 0.0111.

and downregulation of β1-AR in DCs. We sought to determine whether β1-AR signaling suppresses expression of mLT in DCs in vitro. Strikingly, pretreatment of β1-AR agonist (Xamoterol), but not β2- (Procaterol), α1- (Cirazoline), or α2-AR agonist (Medetomidine), suppressed both Mtb- and LPS-induced upregulation of *Lta* mRNA (encodes mLT) in bone-marrow-derived DCs (BMDCs) (Fig. 9a). Then, we tested how β1-AR agonist suppresses Mtb- and LPS-induced *Lta* mRNA upregulation. We found that *Lta* expression requires TRAF3 signaling under toll-like receptor 4 using TRAF3 shRNA (Khaw et al., manuscript in preparation). Therefore, we tested if β1-AR signal affects TRAF3 expression and found that TRAF3 expression was significantly reduced in BMDCs after β1-AR agonist treatment compared to vehicle control (Fig. 9b). These results suggest that the β1-AR signaling directly suppresses mLT in DCs, and that the reduction of β1-AR-mediated suppressive function may trigger mLT upregulation in ELT-EAE mice.

## Discussion

MS clinical pathology and immunological phenotype show a high degree of variation between individuals. Notably, no single pathway, reliable biomarker, or specific treatment has yet been identified for all patients with MS. At present, the response to currently approved therapeutic agents such as interferon-β[43], glatiramer acetate[44], and natalizumab[45] largely varies across the MS population. Mounting evidence suggests that environmental factors play a central role not only in triggering disease onset but also in modifying the course of disease by influencing individual susceptibility and immunological responses, which likely lead to the diverse heterogeneity observed in patients with MS.

Exposure to severe stress and trauma in early life can disrupt various regulatory processes such as the HPA axis[46], autonomic nervous system[47], immune system[19,48,49], and gut microbiome[48,50] across the life span in both animals and humans[51]. A human case study published in 1982 demonstrates that ELT increases MS

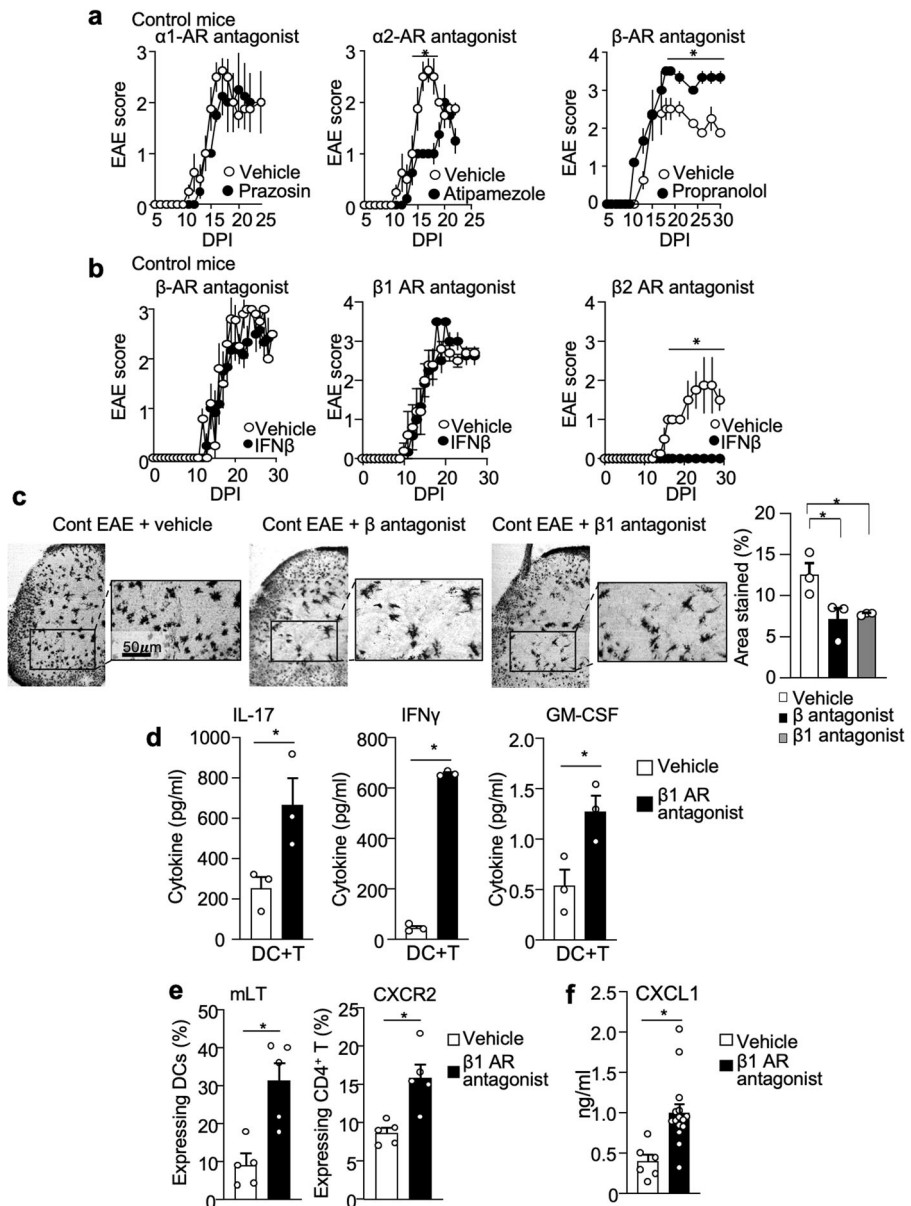

**Fig. 7 β1 adrenergic receptor antagonism is sufficient to reproduce the ELT-induced EAE phenotypes. a** EAE scores of control mice with intraperitoneal injections of α1-AR antagonist (prazosin, 5 mg/kg), α2-AR antagonist (atipamezole, 5 mg/kg), or β-AR antagonist (propranolol, 5 mg/kg) with vehicle control (n = 4 animals/group). All drugs were treated from 0 to 10 dpi every other day. **b** EAE scores of IFNβ-treated control EAE mice subjected to β-AR antagonist (propranolol, 5 mg/kg), β1-AR antagonist (metoprolol tartrate, 5 mg/kg), β2-AR antagonist (ICI 118,55, 1 mg/kg), or vehicle treatment (β-AR antagonist-Veh: n = 5, β-AR antagonist-IFNβ: n = 5, β1-AR antagonist-Veh: n = 5, β1-AR antagonist-IFNβ: n = 6, β2-AR antagonist-Veh: n = 4, β2-AR antagonist-IFNβ: n = 4). **c** Representative images stained quantification of Golgi–Cox staining in the ventral lumbar spinal cord region of vehicle-treated and β1-AR antagonist-treated control EAE mice at 30 dpi. Quantification of area stained by Golgi–Cox (n = 3 animals/group). **d** Amount of cytokines IL-17, interferon γ (IFNγ), and GM-CSF in supernatant derived from culture conditions 72 h post culture initiation. CD4+T cells were isolated from draining lymph nodes of control EAE mice and mice subjected to β1-AR antagonist (n = 3 animals/group). **e** Percentages of mLT-expressing lymph node DCs and CXCR2-expressing CD4+T cells from EAE mice with β1-AR antagonist or vehicle treatment (n = 5 animals/group). **f** Serum levels of CXCL1 at 10 dpi (n = 4 animals/group). Male mice were used for experiments. Each dot represents averaged data per animal. Data are represented as mean ± SEM. Student's t-test, *P < 0.05. Exact P values for asterisks (from left to right): **a** 0.0004; **b** <0.0001; **c** 0.0498; 0.0318; **d** 0.0462, <0.0001, 0.0322; **e** 0.0032, 0.0048; **f** 0.0034.

susceptibility among female patients[52]. More recently, a 2012 study found that patients with MS with histories of physical and/or sexual abuse had significantly higher relapse rates than patients without early-life stress[11]. Here we performed side-by-side comparison of the EAE disease phenotype in adult mice subjected to ELT.

Multiple models of ELT have been proposed, such as maternal separation[53], neonatal handling[54], neonatal saline intraperitoneal injections[55], or a combination of those models[56–58]. Because

emotional neglect and physical abuse affect MS disease[11], we selected an ELT model that involves aspects of both psychological and physical abuse through maternal separation and daily injections, respectively[57]. Differences in observations regarding the effect of ELT on EAE—such as our present finding of the effect of ELT on EAE severity in female mice, contrary to previous results in rats[14]—may be attributed to differences in experimental design or animal species. Further, human case

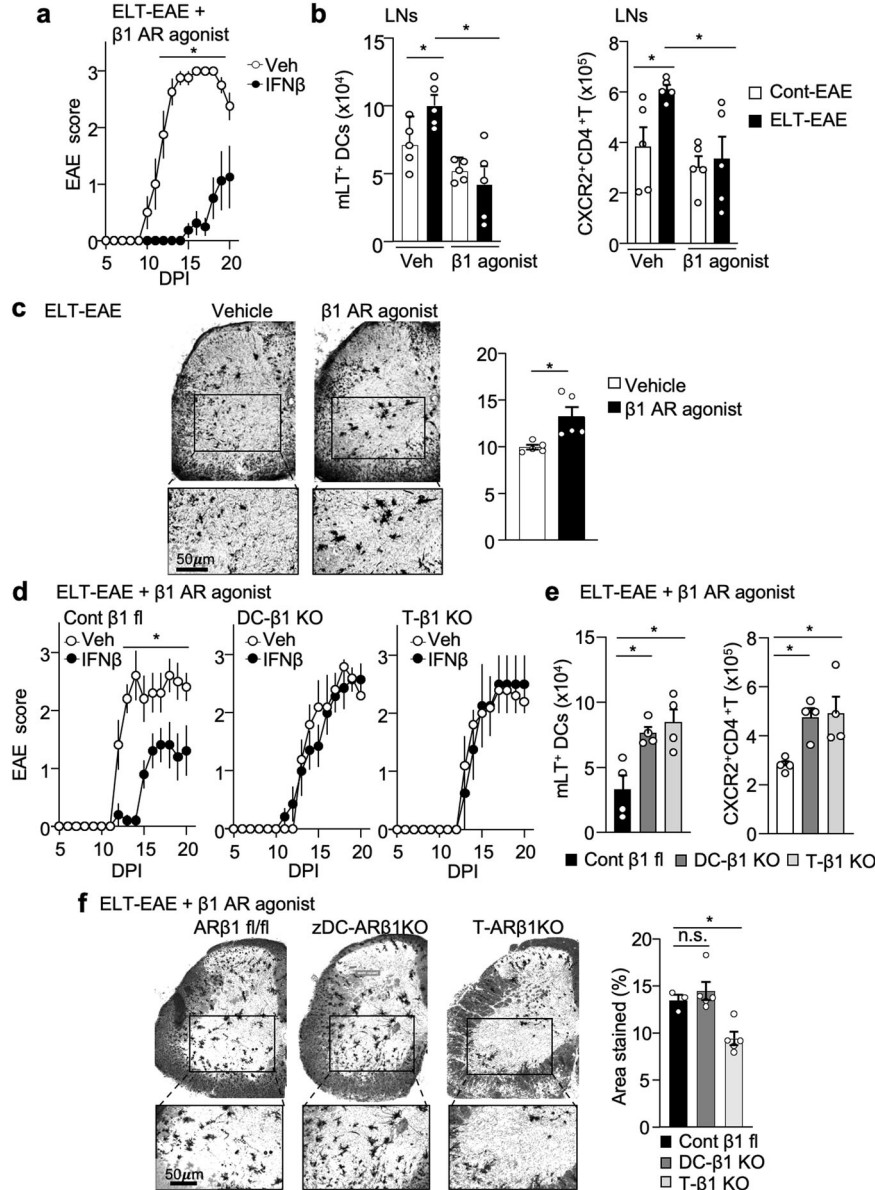

**Fig. 8 β1-AR agonist treatment is sufficient to rescue ELT-EAE subtype. a** EAE scores of IFNβ-treated ELT-EAE mice subjected to β1-AR agonist (Xamoterol, 3 mg/kg) treatment (ELT-β1-AR agonist-Veh: n = 4, ELT-β1-AR agonist-IFNβ: n = 8). Interferon-β was treated from 0 to 9 dpi once every 3 days. Xamoterol was treated from 0 to 10 dpi every other day. **b** Absolute number of mLT-expressing lymph node DCs and CXCR2-expressing CD4+T cells from EAE mice with β1-AR agonist or vehicle treatment (n = 5 animals/group). **c** Golgi–Cox-stained lumbar spinal cord sections of ELT-EAE mice treated either with vehicle or with β1-AR agonist isolated at 30 dpi with quantification of area stained/per field (n = 5 animals/group). **d** EAE scores of IFNβ-treated control β1-AR fl ELT-EAE mice, dendritic cell (DC)-specific β1-AR knockout ELT-EAE mice, and T cell-specific β1-AR knockout ELT-EAE mice subjected to β1-AR agonist (Xamoterol, 3 mg/kg) treatment (ELT-β1-AR agonist-Cont-Veh: n = 5, ELT-β1-AR agonist-Cont-IFNβ: n = 5, ELT-β1-AR agonist-DC-β1KO-Veh: n = 5, ELT-β1-AR agonist-DC-β1KO-IFNβ: n = 7, ELT-β1-AR agonist-T-β1KO-Veh: n = 5, ELT-β1-AR agonist-T-β1KO-IFNβ: n = 5). Interferon-β was treated from 0 to 9 dpi once every 3 days. Xamoterol was treated from 0 to 10 dpi every other day. **e** Absolute number of mLT-expressing lymph node DCs and CXCR2-expressing CD4+T cells from control β1-AR fl ELT-EAE mice, DC-specific β1-AR knockout ELT-EAE mice, and T cell-specific β1-AR knockout ELT-EAE mice subjected to β1-AR agonist (Xamoterol, 3 mg/kg) treatment (n = 4 animals/group). **f** Golgi–Cox-stained lumbar spinal cord sections of β1-AR agonist treated ELT-EAE mice of different transgene knockout conditions at 30 dpi with quantification of area stained/per field (ELT-β1-AR agonist-Cont: n = 3, ELT-β1-AR agonist-DC-β1KO: n = 5, ELT-β1-AR agonist-T-β1KO: n = 5). Male mice were used for experiments. Each dot represents averaged data per animal. Data are represented as mean ± SEM. Student's t-test, *P < 0.05. Exact P values for asterisks (from left to right): **a** 0.0027; **b** 0.0258, 0.0131, 0.0223, 0.0159; **c** 0.0135; **d** 0.0054; **e** 0.0110, 0.0145, 0.0033, 0.0244; and **f** 0.0077.

studies demonstrate that ELT increases MS susceptibility among female patients[52], so our mouse model may be to be more analogous to the human condition. In this study, we determined that EAE induced in mice exposed to ELT has significantly more lasting and rampant inflammatory responses peripherally and in the CNS, which correspond to prolonged severity in observed

hindlimb paralysis and severe neuron damage and demyelination pathologies. Maternal separation alone was sufficient to induce EAE with prolonged severity; however, the application of maternal separation alone is insufficient to produce early disease induction, heightened inflammatory responses, exacerbated demyelination, or neuron damage in the spinal cord. In addition,

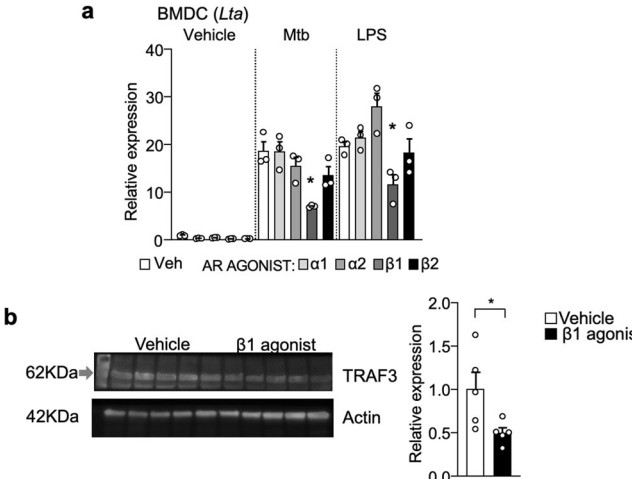

**Fig. 9 β1-AR agonist treatment prevents upregulation of mLT in BMDC.**
**a** Relative expression of *Lta* mRNA in BMDC treated with adrenergic
agonists and/or LPS and Mtb in vitro (*n* = 3 per group). Vehicle or AR
agonists (100 μM) were pretreated to BMDC for 18 h. Then LPS (1 μg/ml)
or heat-killed Mtb (100 μg/ml) was treated for 3 h. **b** Representative
western blot and quantification for TRAF3 in BMDC 18 h after vehicle or β1-
AR agonist (100 μM). *n* = 5 replicates/group. Each dot represents averaged
data per animal. Data are represented as mean ± SEM. Student's *t*-test, *P <
0.05. Exact *P* values for asterisks (from left to right): **a** 0.0042, 0.0249;
**b** 0.0415.

the application of PBS injection alone is insufficient to produce
any altered EAE disease phenotypes in ELT mice. This implies
that the combination of neonatal stressors maternal separation
and sterile PBS injection is essential for induction of entire ELT-
EAE phenotypes.

Elevated invasion of immune cells to the CNS and increased
number of microglia suggest the ability to sustain local CNS
inflammation, mediate neuron damage, and support lymphocyte
survival[59,60]. We also found changes in membrane surface
expression of co-stimulatory molecules CD80 and 4-1BBL on
APCs. APC–T cell interaction through the 4-1BB: 4-1BBL path-
way plays a crucial role in CD4+T and CD8+T cell clonal
expansions[61]. CD80 is a potent co-stimulator of T cell pro-
liferation and generation of cytotoxic lymphocytes[62]. Thus,
upregulation of these co-stimulatory molecules on APCs may
account for increased lymph node and splenic CD4+ and CD8+
T cells observed in ELT-EAE mice.

Strikingly, we discovered that ELT altered responsiveness to
drug treatment during EAE when EAE was induced in adulthood.
Interferon-β sensitivity is a useful marker in distinguishing EAE
phenotype and mechanism. For instance, treatment of interferon-
β given prior to disease onset[27,63,64] suppresses disease severity in
naïve C57BL/6J mice induced with inflammasome-dependent
EAE but not in mice induced with inflammasome-independent
EAE[27]. While EAE controls remained sensitive to interferon-β,
ELT-EAE mice had prolonged interferon-β resistance. Mice
subjected to maternal separation alone also developed interferon-
β-resistant EAE, suggesting that maternal separation may be
crucial in mediating this phenotypic change in ELT-EAE mice.
Consistent with our previous finding[21], interferon-β-resistant
ELT-EAE is independent of inflammasome activity and is largely
driven by the mLT–LTβR axis. Indeed, ELT induces upregulation
of mLT on APCs, and of CXCR2 on CD4+T cells, as well as
upregulation of serum CXCL1, which supports the effector
function of T cell CXCR2. Of note, interferon-β non-responder
patients with MS have been reported to express elevated relative

expression of *CXCR2* in PBMCs compared to interferon-β-
responders[27]. Because CXCL1 had been shown to regulate
NLRP3-inflammasome activation in macrophage via CXCR2
(ref. [65]), the signaling of CXCL1-CXCR2 in myeloid cells of ELT
mice may be contributed to the suppression of inflammasome
activity and IL-1β production in ELT mice, which may provide a
possible mechanistic explanation for interferon-β resistance due
to lack of target of interferon-β[29]. Another possible reason for the
interferon-β-resistant inflammasome-independent feature of
ELT-EAE may be attributed to upregulated lymphotoxin
expression by myeloid cells. Previously we showed that treatment
of recombinant lymphotoxin to control EAE animals were suf-
ficient to render them non-responsive to interferon-β treatment
and induce severe EAE symptoms in inflammasome knockout
($Nlpr3^{-/-}$) animals, suggesting that lymphotoxin may be suffi-
cient to "convert" EAE phenotype from being inflammasome
dependent to being inflammasome independent.

ELT-EAE mice present altered CNS neuron morphology, as
shown in interferon-β-resistant EAE disease[27,64]. The observation
of neuron damage was supported by reduced Golgi–Cox staining
in the spinal cord ventral horn region as well as significant
dendritic spine loss in the same spinal cord region. Dendritic
spine loss has been shown to be associated with motor dysfunc-
tion in MS and EAE[66,67]. Further, we found significantly less
ChAT+ cholinergic motor neurons in ELT-EAE mouse spinal
cords, implying impaired activity of neuron acetylcholine pro-
duction and reduced neurotransmission[68]. Reflective of damaged
neuron status, the combination of ELT and EAE increased
microglial proliferation and activation in the spinal cord. Micro-
glia are readily activated by proinflammatory cytokines IFNγ and
IL-17 under EAE and Alzheimer's disease conditions[69–71].
Because ELT-EAE mice have highly heightened IFNγ-secreting
Th1 cells and IL-17-secreting Th17 cells, it is likely that these
proinflammatory cytokines contribute to microglial activation
under the ELT-EAE condition, leading to CNS neuronal damage.
In addition, because mLT–LTβR signal is reported to equip
CD4+T cells with neurotoxic property changes[27], a large amount
of infiltrated CD4+T cells in the CNS should be also considered to
contribute to CNS neuronal damage in ELT-EAE mice. Further
studies are needed to clarify and deconvolute the exact mechanism
of neurodegeneration in ELT-EAE.

Altered EAE disease phenotype induced by ELT shares
symptomatic and mechanistic similarities with a previously
reported EAE subtype induced by severe activation of innate
immunity using high dosages of adjuvant (Mtb)[27]. This implies
that ELT causes lasting changes to innate immune cell properties,
which result in immune response hyperactivation upon EAE
induction and change the disease phenotype. Because ELT causes
enduring changes in HPA[32] and SNS[31] activity, both of which
regulate immune function, we examined the role of these path-
ways in the observed disease phenotype alteration. Chronic stress
induces HPA activation, as reflected by heightened corticoster-
one. However, we found no significant differences between cor-
ticosterone levels in control EAE and ELT-EAE mice at
adulthood, suggesting that corticosterone may not be an immune-
modulating factor. On the other hand, norepinephrine was
significantly upregulated in both plasma and lymph nodes,
indicative of chronic SNS hyperactivity. This is similar to a pre-
vious study demonstrating that chronic stress results in prolonged
SNS hyperactivity[72]. Notably, heightened norepinephrine levels
were sustained in ELT mice after EAE induction.

Chronic SNS hyperactivity causes downregulation of AR
expression levels in immune cells[35,37]. Accordingly, we found
significantly reduced DC and CD4+T cells that express AR sub-
types (α1, α2, β1, β2) in ELT-EAE mice. Furthermore, cells
derived from ELT mice showed lower amount of β1-AR agonist-

mediated cAMP production which suggest that cells derived from ELT mice express a reduced amount or function of β1-AR. Human and animal studies of MS show evidence of the recovery role of immune cell AR activation[73,74], consistent with anti-inflammatory effects for immune cell ARs. Therefore, the altered EAE phenotype of ELT-EAE mice may be mediated by a hyperactive immune system as a consequence of a reduced regulatory role of the SNS. Importantly, control mice treated with β1-AR antagonist, but not α1-, α2-, or β2-AR antagonist, mimic disease phenotypes of ELT-EAE mice. The finding that β-AR antagonist treatment successfully reproduced severe and prolonged EAE symptoms was unsurprising due to the established immunosuppressive role of β-AR activation on immune cells and a recent article reporting similar results using the EAE model[75]. In contrast, our findings that β1-AR antagonist treatment resulted in interferon-β insensitivity and severe neuron damage are surprising and novel. Further, β1-AR agonist treatment to ELT-EAE mice was sufficient to restore interferon-β sensitivity and lumbar spinal cord neuronal status. This suggests that downregulated β1-AR signaling in immune cells may involve phenotype change upon EAE induction in mice exposed to ELT.

As evidence of dysfunctional immune cell-specific β1-AR being the critical factor for EAE phenotype change, both DC and T cell-specific β1-AR knockout ELT mice did not recover from their interferon-β-resistant status in spite of β1-AR agonist treatment. Furthermore, β1-AR agonist treatment to these specific knockout ELT-EAE animals did not inhibit production of mLT⁺ DCs and CXCR2⁺ CD4⁺T cells, suggesting that β1-AR dysfunction in either DC or CD4⁺T is sufficient to induce these cellular phenotypes. Our present data suggest that β1-AR signaling inhibits mLT transcription by downregulating upstream TRAF3 expression and downregulated mLT expression had previously been associated with reduced CXCR2 expression on CD4⁺T cells, thus immunity of DC-specific β1-AR knockout ELT-EAE mice to β1-AR functional rescue is unsurprising. In contrast, immunity of T cell-specific β1-AR knockout ELT-EAE mice to β1-AR functional rescue was unexpected and suggests that β1-AR signaling in T cells has a regulatory role in mLT expression on DCs via secreted molecule. Additionally, β1-AR agonist treatment rescues neuron damage in DC-specific β1-AR knockout ELT-EAE mice, but not in T cell-specific β1-AR knockout animals. Thus, it is plausible that T cell β1-AR signaling is crucial in regulating T cell neurotoxic effector functions that may be independent of CXCR2. For example, β-AR signaling in T cells have been reported to inhibit production of multiple inflammatory cytokines such as IL-1 and TNF-α which had been shown to perpetuate EAE disease[76,77]. Identification of critical signals for regulating mLT expression and T cell neurotoxic effector are next critical topics.

We have observed significant change in microglia/macrophage population and morphology in the CNS in ELT-EAE mice. Because microglia/macrophages are known to express β-ARs which regulate cell proliferation and cytokine secretion[33,78,79], altered NE levels in ELT mice may also affect β-AR expression levels on microglia/macrophage that may contribute to microglia/macrophage phenotype change and severe disease development as observed in ELT-EAE mice. Previous studies revealed that CNS resident microglia and monocyte-derived macrophages each have distinct roles during EAE development. Specifically, Ransohoff and team[80] reported that monocyte-derived macrophages associate with nodes of Ranvier and initiate demyelination, whereas microglia appear to clear debris. The function of microglia/macrophage and involvement of β-AR signaling for their function in the manifestation of the ELT-EAE disease phenotype will be identified as next topic.

It is known that MS and EAE progression can be determined by DC phenotype and functional properties. In humans, DC properties were shown to vary significantly between patients who suffer from relapse-remitting MS versus progressive MS[81]. In animals, DC-targeting treatments have been shown to dramatically influence EAE outcomes[82]. Our experiment revealed that activation of DC β1-AR signal was sufficient to downregulate expression of mLT, which is an identifying molecule for EAE phenotype alteration in ELT-EAE mice. Because mLT–LTβR signaling is sufficient to induce CXCR2 expression[27], the upregulation of CXCR2, which is another identifying molecule for EAE phenotype alteration observed in ELT-EAE mice, may be mediated by such signaling in ELT-EAE mice. Importantly, we found β1-AR signal in DCs suppresses expression of TRAF3, which is required for *Lta* transcription in DCs (Khaw et al., submitted). Further study to identify the mechanism by which β1-AR signal suppresses TRAF3 expression is our next subject of interest in the effort to identify reliable biomarkers and therapeutic targets for patients with MS who experienced ELT.

In conclusion, our results provide the first evidence that ELT induces not only increased severity and susceptibility in EAE but also phenotypic changes, including altered drug sensitivity, immune cell properties, and CNS neuronal damage. Further, such ELT-mediated altered EAE phenotype may result from dysregulation of immunosuppressive β1-AR signaling in innate immune cells via chronic SNS hyperactivation (summarized in Supplementary Fig. 9). Additionally, we identified mLT, LTβR, and CXCR2 as potential common biomarkers for interferon-β resistance in EAE and MS. Our results also add to emerging evidence that AR agents may be clinically useful as an immunomodulatory therapy for patients with MS who experienced ELT. Further epidemiological studies are needed to expand our understanding of the relationship between chronic stress during childhood and MS phenotype to determine biomarkers and targeted therapeutics for patients with MS who experienced ELT.

## Methods

**Mice.** Healthy C57BL/6J mice aged 6–8 weeks were used in this study. In most experiments, male mice were used. In some experiments, female mice were also used. C57BL/6J mice and TCR2D2 mice (Jackson Laboratory, #6912) were purchased from Jackson Laboratories. Adrb1 fl/fl mice were a gift from Dr. Zigman (University of Texas Southwestern Medical Center) under material transfer agreement (MTA)[83]. DC-specific and T cell-specific Adrb1⁻/⁻ mice were generated by crossing Adrb1 fl/fl mice with Itgax-Cre (Jackson Laboratory 008068) or *Lck*icre (Jackson Laboratory 012837) mice, respectively. Asc⁻/⁻ mice were a gift Genentech under MTA[84]. All mice were kept in group housing (3–5 mice per cage) in a specific pathogen-free facility with a 12-h light/dark cycle and temperature control at 20–22 °C at the Veterinary Medicine Basic Sciences Building at the University of Illinois at Urbana-Champaign. Breeders (one male with one female) were housed in solid-bottom caging with standard bedding. Rodent diet (Teklad) and water were provided ad libtum. Mouse litter size and sex ratio varies between breeders and pregnancies. Male mice were used for experiments unless otherwise mentioned. The study was approved by the University of Illinois Institutional Animal Care and Use Committee (protocol number 19171).

**ELT model.** We created an ELT model by subjecting mouse pups to maternal separation with PBS injections. First, mouse pups were separated daily from their mother and father (3–6 months old) for 3 h (3–6 pm) during postnatal days 2–14. Newborn mice share their cage with parents until wean time. Neonates subjected to ELT or no neonatal treatment were derived from different litters. To induce physical stress, intraperitoneal injections of 40 μL PBS were administered to pups with a 27G needle during postnatal days 4–10 while they were separated from their mother as an additional stressor. During separation, pups were kept together in a dark covered cage with supplemental heat to avoid pup abandonment due to low body temperature. Normal parental care-taking behavior was observed after reunification despite the intervention. Pups were weaned from their mothers on postnatal day 28.

**Induction of active EAE.** As a control, we used age- and sex-matched C57BL/6J mice weaned at postnatal day 28. In 6–8-week-old mice, we induced EAE by subcutaneous injection of myelin oligodendrocyte glycoprotein (MOG)₃₅₋₅₅ peptide (100 μg/mouse) emulsified in complete Freund's adjuvant including heat-killed Mycobacteria (Mtb, BD Difco, DF3114-33-8, 200 μg/mouse) on day 0, and intraperitoneal injection of pertussis toxin (200 ng/mouse) on days 0 and 2. We evaluated EAE score (1: tail limpness; 1.5: reversible impaired righting reflex; 2:

impaired righting reflex; 2.5: one hindlimb paralysis; 3: both hindlimb paralysis; 3.5: both hind and one forelimb paralysis; 4: hind and forelimb paralysis; 5: death) for 30 days and compared EAE severity and duration between ELT and control mice. We provided water gel and powder feed for mice that scored ≥2 to avoid body weight reduction as a result of being unable to reach standard food and water. Disease score analysis was performed in the middle of the day during the light cycle. For optimize EAE disease severity, we also induced EAE using Mtb (50 or 100 μg/mouse) under same protocol as mentioned above. Successfully induced mice achieved an EAE score of ≥1.

**Materials**. All information oregarding reagents, antibodies, and ELISA kits are listed in Supplementary Table 1.

**Pharmaceutical treatment of mice**. All treatments were conducted through i.p. injections. Day 0 represents day of EAE induction. Interferon β ($3 \times 10^4$ unit/mouse) was i.p. injected into mouse every other day or three day from days 0 to 9 as previously performed[27,29]. CXCR2 inhibitor (SB225002, 0.1 mg/kg mouse) was i.p. injected into mouse every day from day 0 to day 20 (ref. [27]). α1-Selective antagonist (prazosin, 3 mg/kg)[38], α2-selective antagonist (atipamezole, 3 mg/kg)[39], β1-selective antagonist (metoprolol, 3 mg/kg)[40], and β2-selective antagonist (ICI 118,551, 0.5 mg/kg)[41], β-adrenergic receptor non-selective antagonist (propranorol, 5 mg/kg), and β1 agonist (xamoterol, 3 mg/kg)[42] were i.p. injected every other day from day 0 to day 10. As control, vehicle (PBS) for interferon β, SB225002, atipamezole, metoprolol, ICI 118,551, propranorol, and xamoterol, and vehicle (3% DMSO) for prazosin were treated with same protocols.

**Immunohistochemistry**. Spinal cords were harvested from PBS-perfused and 4% paraformaldehyde-fixed mice at 30 dpi. Spinal cords were post-fixed in 4% paraformaldehyde overnight and then cryoprotected by immersion in 30% sucrose solution for 24 h. Samples were frozen in OCT compound and stored at −80 °C until cryostat sectioning. Transverse sections (30 μm) of spinal cords were mounted on poly-L-lysine-coated glass slides. To visualize microglia and cholinergic neurons, mounted samples were permeabilized with 0.05% Triton-X for 15 min at room temperature, blocked with 2% BSA for 2 h at room temperature, incubated overnight at 4 °C with goat polyclonal AIF-1/Iba1 primary antibody or cholinergic acetyltransferase antibody diluted in PBS, and incubated with chicken anti-goat Alexa 647 secondary antibody or chicken anti-goat Alexa 488 secondary antibody for 2 h. Labeled samples were dried, covered with mounting media (Prolong Gold Antifade Mountant, Invitrogen, Cat# P36930), and sealed with a coverslip. For Iba1 analysis, tissue sections (at least three images of ventral roots from individual L4–L6 lumbar spinal cord sections per animal) were visualized using a Nikon A1 confocal scanning microscope at ×20 magnification. A total of 614 Iba1+ cells (in three animals per condition) were included in our analyses of soma size and roundness using the ImageJ *morpholibj* plugin[85]. Briefly, A region of interest (ROI) was manually placed over spinal cord areas. Gray scale attribute filtering was applied with Operation set to "Opening", Attribute set to "Area", Minimum value set to "25 pixels", and Connectivity set to "8". Morphological filter was subsequently applied with Operations set to "Opening", Element set to "Octagon", and Radius (in pixels) set to "1". A standardized threshold was applied across all ROIs. Measurements of "soma area" and "roundness" were obtained using "analyze particles" with measurements set to include "area, shape description, fit eclipse". For ChAT analysis, tissue sections (at least three images of ventral roots from individual L4–L6 lumbar spinal cord sections per animal) were visualized using a Nikon A1 confocal scanning microscope at ×20 magnification. Neurons labeled with ChAT were counted manually by a single blinded experimenter.

**Luxol-fast blue staining and analysis**. Transverse sections (30 μm) of frozen spinal cord from mice at 30 dpi mounted onto poly-L-lysine-coated slides and left to dry overnight. Sections were hydrated with 95% ethyl alcohol, and then placed in 1:1 ethanol/chloroform for 5 min. Sections were rehydrated with 95% ethyl alcohol then incubated in 0.1% Luxol-fast blue solution for 12 hours. After 12 h, 95% ethanol was used to wash away excess dye solution, followed by rinsing step with distilled water. Li$_2$CO$_3$ (0.05%) solution was applied to sections for 30 s, followed by a wash step with 70% ethanol until a sharp contrast between gray and white matter becomes apparent. Sections were immersed in 90, 95, and 100% ethanol gradient dehydration washes, followed by the application of a coverslip. For image analysis, gray scale images were opened in ImageJ and an automatic threshold was applied. Region of interest was manually drawn to restrict analysis to the area of white matter regions. Analyses were performed on at least five serial sections. Quantitative analyses measured the area of Luxol-fast blue staining and normalized this area to the total analysis area.

**FluoroJade-C staining**. Transverse sections (30 μm) of frozen spinal cord from mice at 30 dpi mounted onto poly-L-lysine-coated slides and left to dry overnight. Sections were immersed in 1% NaOH/80% EtOH for 5 min followed by one 2-min rinse with 70% EtOH and one 2-min rinse with distilled water. Sections were incubated in 0.06% KMgO$_4$ solution for 10 min and rinsed with distilled water for 2 min. Sections were then immersed in 0.0001% FluoroJade solution for 10 min. After 10 min, sections were rinsed with distilled water for 1 min, repeated three

times. Slides were dried at 50 °C for 5 min followed by immersion in xylene for 1 min. Promount and coverslip were applied to dried slides for extended storage. For image analysis, ventral gray matter region of lumbar spinal cord sections were visualized using a Nikon A1 confocal scanning microscope at ×20 magnification. The percent of stained area was averaged across serial sections per animal. Positively stained neurons from at least five images of ventral roots from individual L4–L6 lumbar spinal cord sections per animal were counted manually by a single blinded experimenter. Count value was normalized by image area.

**Golgi–Cox staining**. Mice were deeply anesthetized with isoflurane and perfused intra-cardially with PBS (15 ml) followed by 4% paraformaldehyde (pH 7.4) (25 ml) at 30 dpi. Spinal cord was removed and transferred into solutions designation by the FD Rapid Golgi-Stain Kit (FD Neuro-Technologies, Inc.). The protocol is described as follows: organs were placed in 10 ml of Solution A and B of ratio 1:1 for 24 h and then immersed in an identical volume of fresh Solution A and B for an additional 14 days in dark. Spinal cord and brain were then transferred to solution C for 3 days, with a replacement of fresh solution after 24 h. To attain cryoprotection, spinal cords were then transferred in a 30% sucrose solution overnight or until organs sunken to bottom. Spinal cords were individually snap frozen in OCT on dry ice and kept frozen at −80 °C to be cut into 50-μm-thick coronal slices in Cryo-cut 1800 (Reichert Jung) at −14 °C. Slices were then incubated in solution D and E before being mounted on a glass slide, dehydrated, and sealed by prolong gold mountant. Samples were visualized using Leica DM4000 bright field modality equipped with ×20 objective. Five fields of spinal cord ventral gray matter were analyzed per animal to obtain average value per animal. For image analysis, gray scale images were opened in ImageJ and an automatic threshold was applied. Wand tool was used to distinguish white from gray matter to identify ventral gray matter region as of interest. Percentage stained with Golgi–Cox was determined as "Area of thresholded positive staining/Total ROI area".

**Confocal reflection super-resolution acquisition**. Samples of Golgi–Cox-stained spinal cords were imaged using a Nikon A1 confocal scanning microscope under the confocal modality and CRSR modality (with minimized pinhole at 0.3 AU) using ×100/1.49 NA oil objective. A 405-nm continuous wave laser was used, and reflectance mirror (BS 20/80) was applied. For dendritic spine analysis, z-stacks of at least 100 intervals were acquired. Pixel dimensions are: x: 0.0628 μm; y: 0.0628 μm; z: 0.075 μm. Four to eight z-stacks of spinal cord ventral roots from four to eight individual 50-μm-thick spinal cord sections per animal were visualized. Twenty dendrites per animal were included in our analyses using filament tracer autopath function (Imaris)[24]. Briefly, dendritic regions that was 40–60 μm in length and void of dendritic branch points and crossing neurites were selected. Gaussian filter and background subtraction were applied to z-stacks of cropped individual dendrites prior to tracing filaments.

**Flow cytometry**. Spleen, lymph nodes (axillary and inguinal), brain, and spinal cord were removed, and then minced with a sterile razor blade. Minced spleen and lymph nodes were passed through a mesh filter. Brain and spinal cord were digested in collagenase for 30 min at 37 °C, and then passed through a 70-μm filter to remove debris. CNS-infiltrating lymphocytes were isolated by percoll gradient centrifugation. After washing, cells were suspended in ice-cold staining buffer (sterile PBS containing 2% FBS). Cells were blocked with anti-CD16/32 on ice for 10 min. After blocking, $1 \times 10^6$ cells were suspended in 0.1 ml of ice-cold staining antibody buffer and stained on ice in the dark for 20 min. After washing twice in staining buffer, samples were acquired on a Cytek Aurora flow cytometer. Gates were determined using unstained and single-stained samples obtained from the same tissue of origin. Results were analyzed using FCS Express 6 flow cytometry software (De Novo Software).

**In vitro cell culture of ex vivo lymph node immune cells**. DCs (CD11c+) were isolated from inguinal and axillary lymph nodes of EAE mice with or without ELT condition (both 6-week-old) or EAE mice with vehicle or β1-AR antagonist treatment by beads positive selection (EasySep Mouse Biotin Positive Selection kit) with Biotinylated CD11c antibody. CD4+ T cells were also isolated from inguinal and axillary lymph nodes of naïve TCR2D2 mice, a myelin oligodendrocyte glycoprotein (MOG)-specific transgenic T cell mouse, EAE mice with or without ELT condition, or EAE mice with vehicle or β1-AR antagonist treatment by beads negative selection (EasySep Mouse CD4+ T cell isolation kit). CD4+ T cells were labeled by CellTrace Violet Cell proliferation kit or CFSE Cell proliferation kit according to manufacturer's protocol. After washing, CD4+ T cells were co-cultured with DCs (each $10^5$ cells) and treated with MOG35-55 peptide (5 μg/ml) in complete RPMI 1640 medium for 72 h in 96-well round plates. Cells derived frominguinal and axillary lymph nodes of mice with or without ELT condition were also labeled by CFSE. After washing, lymph node-derived cells were also treated with MOG35-55 peptide (5 μg/ml) in complete RPMI 1640 medium for 72 h. We analyzed CD4+ T cell proliferation by evaluating fluorescence in CD4+ T (CD3+CD4+) cells by flow cytometry (Cytek Aurora, UIUC).

**Enzyme-linked immunosorbent assay (ELISA) analysis**. Blood samples were taken at approximately 9 a.m. from all mice to ensure consistency in sample quality. Serum or plasma were isolated and stored at 80 °C until ELISA analysis. ELISA was conducted according to manufacturer's protocol. Triplicate wells were made for each sample derived from one mouse. Sample diluent alone was used as a negative control.

**RNA and cDNA preparation for standard qPCR analyses**. For standard qPCR analysis, DCs (CD11c$^+$) were isolated from inguinal and axillary lymph nodes by beads positive selection (EasySep Mouse Biotin Positive Selection kit) with Biotinylated CD11c antibody. Macrophages were isolated from lymph nodes of ELT-EAE and control EAE mice at EAE 10 days by beads separation (EasySep Mouse CD11b Positive Selection Kit II, STEMCELL) after removing neutrophil population by beads selection (STEMCELL). Total RNA was extracted from cells with RNeasy Kit (Qiagen). cDNA synthesis was performed with qScript cDNA SuperMix (Quanta). qPCR analysis was performed with KAPA-SYBR-FAST (KAPA Bio Systems) with an initial denaturing step at 95 °C for 3 min, followed by 35 cycles of a denaturation step at 94 °C for 3 s and an annealing and extension step at 60 °C for 30 s. Relative amounts of qPCR products were determined with the $\Delta\Delta Ct$ method to compare the relative expression of target genes and beta-actin housekeeping genes. The expression of the gene encoding β-actin was used as an internal control. Primer information are listed in Supplementary Table 2.

**DNA methylation assay**. DCs (CD11c$^+$) were isolated from inguinal and axillary lymph nodes by beads positive selection (EasySep Mouse Biotin Positive Selection kit) with Biotinylated CD11c antibody from spleen of control EAE and ELT-EAE mice at EAE 9-dpi. Genomic DNA was obtained with the GenElute$^{TM}$ Mammalian Genomic DNA Miniprep kit (Sigma)[27]. Bisulfite conversion was performed with the MethylDetector kit (Active Motif), and the *Lta* promoter (between −363-nt and +65-nt from the transcription start site) was cloned into the pCR4-TOPO-TA vector (Life Technologies) and transformed into *E. coli*. Methylation status was evaluated by sequencing 17–22 clones per group. Percentages of unmethylated CpG in all the tested CpG sites were calculated.

**In vitro-stimulated BMDCs**. Bone marrow was isolated from naïve C57BL/6J mice. Isolated cells were treated with recombinant GM-CSF (200 ng/ml) for 6 days to generate BMDCs. For qPCR experiment, BMDC (10$^6$ cells) were treated with AR agonists (10 μg/ml) for 18 h at 37 °C according to a previous report, then treated with heat-killed Mtb (100 μg/ml) or lipopolysaccharides (1 μg/ml) for 3 h at 37 °C. As control, BMDC were treated with vehicle (PBS) instead of AR agonists, and vehicle instead of Mtb or LPS. For western blotting experiments, BMDC (5 × 10$^6$ cells) were used under the same treatment condition, respectively. To detect TRAF3 and actin, we analyzed BMDC lysates by western blotting with antibody against TRAF3 (1:1000) and actin (1:1000). To normalize TRAF3 signals by beta-actin in the same blotting membrane, we cut membrane and stained them with specific antibodies. Horseradish peroxidase (HRP)-conjugated secondary antibody (1:1000) was used to detect signals, which were visualized by enhanced chemiluminescence (Peace). Images were taken by FluorChem Systems (Bioteche). Quantification of image was performed by ImageJ.

**Statistical analysis**. Statistical analysis in all results was evaluated with two-tailed unpaired Student's *t*-tests and *t*-values. The criterion of significance was set as $P < 0.05$. Animals were randomly used for experiments under the criteria aforementioned in the section of "Animals." All behavior experiments were performed in a blinded and randomized fashion. No animals or data points were excluded. Quantifications were performed from at least two experimental groups in a blinded fashion. Measurements were taken from distinct samples for each experiment. For each test, the experimental unit was an individual animal. No statistical methods were used to predetermine sample sizes, but our sample sizes are similar to those generally employed in the field. Data distribution was assumed to be normal, but this was not formally tested.

**Reporting summary**. Further information on research design is available in the Nature Research Reporting Summary linked to this article.

## Data availability
The data supporting the findings of this study are available within the article and its Supplementary files or from the corresponding author upon reasonable request. Raw data are provided in a Source Data file.

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

## Acknowledgements
We thank Mary Clutter for helping with sample isolations and analysis. We also thank Dr. Jeffrey Zigman (UT Southwestern) who provided us β1-AR floxed mice. This research was supported by University of Illinois start-up funds and Sumitomo Foundation (MI).

## Author contributions
Y.M.K. and M.I. designed the study, analyzed data, and wrote the manuscript. D.M. performed the methylation study. S.O. performed the Luxol-fast blue and Fluor Jade C studies. E.K. performed sample preparations for flow an IHC analyses.

## Competing interests
The authors declare no competing interests.
