## [Peer Review File · Nature Communications]

Reviewers' comments:

Reviewer #1, expert in neuroimmunology (Remarks to the Author):

In this manuscript, Khaw and colleagues show that early-life-trauma (ELT) leads to an EAE phenotype that is resistant to interferon-beta treatment. The EAE phenotype is also characterized by the absence of remission and increased neuronal loss. The authors argue that reduced B1AR signaling in immune cells after ELT explains the described EAE phenotype in adult mice. The manuscript is interesting and deals with an important issue, however it contains many flaws that preclude acceptance in its present form.

Major concerns

1 – In Figure 2 when the authors show data related to the immune response in the periphery, there is only information about the absolute numbers of different cell populations. Nothing about the antigen-specific T cell response in the draining lymph nodes after immunization. The authors should show T cell proliferation and cytokine (IFN-gamma, IL-17, GM-CSF, IL-2) secretion upon antigen-specific stimulation comparing control and ELT mice. In figure 2C, the authors show an increased frequency of cells (DCs) expressing CD80 and 4-1BBL. In Supplementary figure 2 the authors show data claiming that maybe because of that, DCs coming from ELT mice induce higher proliferation of 2D2 cells. Usually, it is very easy to see the dilution peaks of the proliferation dye identifying the generations of the dividing 2D2 cells. That is not the case for the histogram in Supplementary figure 2D. Authors should show a figure overlaying histograms from proliferation assays using DC coming from control and ELT mice. In figure 2 D and E, the authors show increased absolute numbers of T cells and IL-17- and IFN-g- secreting T cells taken from the spinal cord of ELT mice (as compared with controls) 18 dpi. However, it seems that at this time point there is little, if any, difference in disease score comparing control and ELT mice. The differences become clear later on, after 20 dpi, mainly because the absence of remission in ELT mice. How do the authors discuss this issue? Wouldn't be better to look at cell infiltration in the CNS later on?

2- In Figure 6 the authors show data claiming that ELT downregulates adrenergic signaling in immune cells, particularly in DCs and CD4+ T cells. However, they show number of cells expressing the adrenergic receptors. They do not show the level of expression of the receptors in immune cells or the signaling response after the activation of adrenergic receptors in immune cells when comparing control and ELT mice. Do DCs and CD4+ T cells coming from control and ELT mice display any differences in cAMP levels or CREB phosphorylation upon B1AR activation?

3- Figure 7. Does B1AR antagonist treatment interfere with the generation of the antigen specific encephalitogenic T cell response? Does BA1R antagonist treatment increase CXCL1 secretion (as shown in Figure 5G)? Does B1AR antagonist treatment increase CXCR2 expression in CD4+ T cells?

4- Figure 8. Does B1AR agonist treatment revert the EAE phenotype in ELT mice (not only susceptibility to interferon beta treatment)? Does it avoid increased expression of mLT in DCs and CXCR2 in CD4 T cells? The authors argue that downregulation of B1AR signaling in immune cells explains EAE phenotype in ELT mice, however there is no in vivo data giving support to that claim. Does B1AR agonist treatment in animals lacking B1AR only in hematopoietic cells fail to revert the EAE phenotype in ELT mice? A chimera experiment can answer this important question; giving support to the claim that downregulation of B1AR signaling in immune cells is directly related to EAE phenotype in ELT mice.

Minor Concerns

1 – In Figure 1d is very hard to notice the differences in LFB staining of spinal cord sections. It would be recommended to replace the images or to show higher magnification pictures.

2- The authors show a lot of data from flow cytometry experiments, however the plots and gate strategy are not shown anywhere. It would be advisable to show that, for instance, for data in Figure 2, figure 5f, figure 6c and 6d, figure 7c

3 - There is data in the literature showing that CXCL1 and CXCL2 Regulate NLRP3 Inflammasome activation via CXCR2 (J Immunol September 1, 2017, 199 (5) 1660-1671). Curiously, it seems that EAE-ELT mice display reduced IL-1 β levels and that this EAE phenotype is ASC independent. What would be the authors comment on that? In addition, do myeloid cells also display increased CXCR2 expression in EAE-ELT mice?

Reviewer #2, expert on trauma and stress responses (Remarks to the Author):

Khaw and colleagues studied the influence of early life trauma (ELT) on experimental immune encephalitis (EAE) phenotype in mice. The study suggests immune cell β 1 adrenergic signaling as a druggable molecular target to interfere with ELT-EAE phenotype.

Not surprisingly, there is only limited knowledge about the relation between ELT and MS in humans, since approaches to study this relation would require a sound understanding in how far MS patients suffered from ELT – such an approach appears virtually impossible to conduct, not only because of practical and ethical concerns. The study at hand will broaden the view concerning the heterogeneity of MS and its relationship with an additional environmental factor (ELT). Clearly, this is of importance, as it may help to increase the quality of MS treatment and to assess disease progression. The work might give interesting implications for an option treating MS patients, who do not respond to IFN β therapy. Furthermore, the paper suggests biomarkers that appear promising for MS studies in humans.

The manuscript is well written, and the authors have undertaken a substantial amount of work to provide evidence for their findings. The presentation of the results is clear and comprehensible. The authors discussed their findings in the context of the existing and relevant literature.

The authors may wish to consider the following points to further improve their manuscript:

The authors used a model to create ELT by 'subjecting mouse pups to maternal separation with phosphate buffered saline injections'. Please explain the usage of PBS injection additional to maternal separation. I also did not understand the rationale of the sentence 'First, mouse pups were separated daily from their mother and father.' Do the authors normally breed newborn mice together with the father?

I am wondering that only one control group was used for the data presented in Figures 1-3. There should be three control groups, since it remains questionable, whether the measured effect (immune phenotype) results from the combination of maternal deprivation and injection (as stated by the authors), or whether it is simply the result of the injection alone. The authors should use the following three control groups: i) no handling, ii) maternal deprivation, iii) ip. injection alone.

Since many factors are influencing the variability in ELS animal models (summarized in PMC6079200) the method section should be further specified. Detailed information according ARRIVE criteria are missing (e.g. clear background information of the used mouse strains: C57B6N or C57B6J, number of animals, litter size, sex ratio per litter). Was the approach "control vs ELT treatment" conducted in the same litter or different litters? Were the animals in Fig 5-8 male or female mice?

Additional methodological information should be supplied, e.g. for the PCR method. Which primer sequences and housekeeping genes have been used? Which vehicle substance was used for the pharmaceutical treatment of AR agonist and antagonist?

Statistics: The authors assume a normal distribution of the data though they did not test for normal distribution. Since the authors present SEM instead of SD, the exact n for each experimental group must be indicated in the figure legends. In addition, the text in the section "statistical analysis" is identical with the text in another paper – Please reword to avoid self-plagiarism.

Figure 3g: IBA1 stains both resident microglia and blood derived macrophages, which invade the brain during EAE. Both populations differ in phenotype and function. As shown in Figure 2f, the authors distinguished between macrophages and microglia in the EAE affected spinal cord. Microglia / macrophages express adrenoceptors and could respond to altered NE levels due to SNS hyperactivity. Is there anything known about the distinct roles and activation pattern of this two populations in the EAE affected tissue? Could microglia / macrophages be the link between ELT-EAE and increased severity and susceptibility in EAE? Please comment.

Figure 6 shows plasma and lymph node NE levels in adult control and ELT mice. Is there any information about peripheral and central NE levels in control and ELT mice after EAE induction? I wonder why the authors stopped measuring before EAE induction? Please comment.

Figure 4b: Do female control EAE mice show the same EAE score pattern than male control EAE mice with and without IFN β treatment?

Suppl Fig 5: Serum corticosterone levels did not differ between control and ELT group in EAE mice. Since corticosterone release follows a circadian rhythm, the time of sampling influences the results. Please state whether the samples were taken at the beginning or the end of the activity phase of the mice.

Minor points:

The rationale for using ASC $^{-/-}$ mice is unclear. Please add an appropriate introduction of this approach and explain the concept of inflammasome dependent and independent EAE in more detail.

Abstract: "...however, the underlying mechanism is yet well defined." Probably the authors want to express "...the underlying mechanism is not yet well defined. "?"

AUTHOR RESPONSE TO REVIEWERS' COMMENTS

**Reviewer #1, expert in neuroimmunology (Remarks to the Author):**

*In this manuscript, Khaw and colleagues show that early-life-trauma (ELT) leads to an EAE*
*phenotype that is resistant to interferon-beta treatment. The EAE phenotype is also*
*characterized by the absence of remission and increased neuronal loss. The authors argue that*
*reduced B1AR signaling in immune cells after ELT explains the described EAE phenotype in*
*adult mice. The manuscript is interesting and deals with an important issue, however it contains*
*many flaws that preclude acceptance in its present form.*

We thank the reviewer 1 for giving us many valuable comments to further improve our
manuscripts. As mentioned below, we have conducted all suggested experiments. It is our belief
that the manuscript is substantially improved after making the suggested edits.

**Major concerns**

**REVIEWER COMMENT:**

*1 – In Figure 2 when the authors show data related to the immune response in the periphery,*
*there is only information about the absolute numbers of different cell populations. Nothing about*
*the antigen-specific T cell response in the draining lymph nodes after immunization. The authors*
*should show T cell proliferation and cytokine (IFN-gamma, IL-17, GM-CSF, IL-2) secretion upon*
*antigen-specific stimulation comparing control and ELT mice. In figure 2C, the authors show an*
*increased frequency of cells (DCs) expressing CD80 and 4-1BBL. In Supplementary figure 2 the*
*authors show data claiming that maybe because of that, DCs coming from ELT mice induce*
*higher proliferation of 2D2 cells. Usually, it is very easy to see the dilution peaks of the*
*proliferation dye identifying the generations of the dividing 2D2 cells. That is not the case for the*
*histogram in Supplementary figure 2D. Authors should show a figure overlaying histograms from*
*proliferation assays using DC coming from control and ELT mice. In figure 2 D and E, the*
*authors show increased absolute numbers of T cells and IL-17- and IFN-g- secreting T cells*
*taken from the spinal cord of ELT mice (as compared with controls) 18 dpi. However, it seems*
*that at this time point there is little, if any, difference in disease score comparing control and ELT*
*mice. The differences become clear later on, after 20 dpi, mainly because the absence of*
*remission in ELT mice. How do the authors discuss this issue? Wouldn't be better to look at cell*
*infiltration in the CNS later on?*

**AUTHOR RESPONSE:**

We thank the reviewer for all his/her critical comments on immune cell properties in periphery
and CNS. Regarding antigen-specific T cell response in draining lymph nodes after
immunization, we carried out experiments to examine T cell proliferation and cytokine secretion
upon antigen-specific stimulation using lymph node CD4⁺T cells and co-culture condition of
CD4⁺T cell and DC isolated from draining lymph nodes, respectively, according to reviewer's
suggestion. We found that antigen-stimulated T cell proliferation and cytokines (IFN-gamma, IL-
17, GM-CSF, but not IL-2) production are significantly higher in ELT-EAE condition than that in

control-EAE condition. We added these data in the revised manuscript as Figure 2d, Figure SI
2j, and Figure SI2h.

Additionally, we agree with the reviewer that we should show a figure of overlaying
histograms to display T cell proliferation using proliferation dye. We added such new figure as
SI Figure 2i.

We thank reviewer for pointing out the investigating infiltrated T cell number at disease
late time point. We agree with the reviewer's rationale that because disease scores between
control EAE and ELT-EAE groups are similar at 18 dpi, it would be more relevant to investigate
T cell infiltration to the CNS at a later time point when disease scores significantly vary between
groups. To that end, we have analyzed CD4⁺ T cell count and properties at 25 dpi, which is
when distinct difference disease score between control and ELT mice are observed (Fig.1 b).
Similar to data obtained from 18 dpi, we found higher number of CD4⁺ T cells, Th17, and Th1
cells in ELT-EAE mice at 25 dpi. We added these data as Figures 2e, f, and g in the revised
manuscript.

**REVIEWER COMMENT:**

*2- In Figure 6 the authors show data claiming that ELT downregulates adrenergic signaling in*
*immune cells, particularly in DCs and CD4+ T cells. However, they show number of cells*
*expressing the adrenergic receptors. They do not show the level of expression of the receptors*
*in immune cells or the signaling response after the activation of adrenergic receptors in immune*
*cells when comparing control and ELT mice. Do DCs and CD4+ T cells coming from control and*
*ELT mice display any differences in cAMP levels or CREB phosphorylation upon B1AR*
*activation?*

**AUTHOR RESPONSE:**

We thank the reviewer for the valuable comment about demonstrating the level of AR receptor
activation signal in DC and T cells, which allows us to provide additional support to our data
interpretation. According to reviewer's suggestion, we have measured cAMP levels upon B1AR
agonist treatment to DC and CD4⁺T cells isolated from control and ELT mice. Similar to B1AR
expression, cells derived from ELT mice showed lower amount of B1AR agonist-mediated
cAMP production, compared with cells derived from control mice, suggesting that cells derived
from ELT mice have reduced B1AR signaling. New data are presented as Figure 6e and f in the
revised manuscript.

**REVIEWER COMMENT**

*3- Figure 7. Does B1AR antagonist treatment interfere with the generation of the antigen*
*specific encephalitogenic T cell response? Does BA1R antagonist treatment increase CXCL1*
*secretion (as shown in Figure 5G)? Does B1AR antagonist treatment increase CXCR2*
*expression in CD4+ T cells?*

**AUTHOR RESPONSE:**

We thank the reviewer for all comments on B1 AR antagonist effect. We have conducted
appropriate experiments to answer the reviewer's questions. We found the significantly higher
amount of antigen-induced IL-17, IFN γ , and GM-CSF cytokines in co-culture system between

DC and CD4⁺T cells derived from B1AR antagonist-treated control mice than vehicle-treated
mice (Figure 7d). In addition, we also found higher levels of CXCL1 production in the serum
(Figure 7f), mLT expression on DC (Figure 7e), and CXCR2 expression on CD4⁺T cells (Figure
7e) in B1AR antagonist-treated control mice when compared with vehicle-treated mice. All these
data suggest that treatment of B1AR antagonist is sufficient to mimic properties of ELT mice
with B1AR downregulation in the context of EAE. All data are presented in the revised
manuscript.

**REVIEWER COMMENT:**

*4- Figure 8. Does B1AR agonist treatment revert the EAE phenotype in ELT mice (not only*
*susceptibility to interferon beta treatment)? Does it avoid increased expression of mLT in DCs*
*and CXCR2 in CD4 T cells? The authors argue that downregulation of B1AR signaling in*
*immune cells explains EAE phenotype in ELT mice, however there is no in vivo data giving*
*support to that claim. Does B1AR agonist treatment in animals lacking B1AR only in*
*hematopoietic cells fail to revert the EAE phenotype in ELT mice? A chimera experiment can*
*answer this important question; giving support to the claim that downregulation of B1AR*
*signaling in immune cells is directly related to EAE phenotype in ELT mice.*

**AUTHOR RESPONSE:**

We thank the reviewer for all comments on rescue studies with B1 AR agonist. We have
addressed by conducting relevant experiments for all comments. We show in new Figure 8b
that B1AR agonist treatment to ELT-EAE mice was sufficient to induce rescue in expression
levels of mLT in DCs and CXCR2 in CD4⁺ T cells.

Regarding the involvement of B1AR in immune cells on ELT phenotype, we have
conducted several experiments. As suggested by reviewer, we tried to do chimera experiments
using B1 AR KO mice. However, because cryorecovery step is required to obtain B1 AR KO
mice from Jackson laboratory, which greatly hinders our timeline to complete this critical study
within the limited time period, we decided to address the issue using a different way. By
collaborating with Dr. Jeffrey Zigman (UT Southwestern) whom we acknowledged in the
manuscript, we successfully made DC-specific and T-cell specific B1AR KO mice and
conducted the following experiments using these mice. We evaluated IFN β sensitivity,
expression levels of mLT in DCs and CXCR2 in CD4⁺ T cells, and CNS neuronal damage in
ELT-subjected DC-specific and T cell-specific B1AR KO mice with B1AR agonist treatment, and
found that rescue functions by B1AR agonist are canceled in DC-specific and/or T-cell specific
B1AR KO mice. Therefore, we successfully demonstrated the involvement of B1AR in immune
cells for ELT phenotype. New data are shown in Figure 8d, e, and f in the revised manuscript.

**Minor Concerns**

**REVIEWER COMMENT:**

*1 – In Figure 1d is very hard to notice the differences in LFB staining of spinal cord sections. It*
*would be recommended to replace the images or to show higher magnification pictures.*

**AUTHOR RESPONSE:**

We thank the reviewer for this comment. We have replaced panels with images of higher
resolution.

**REVIEWER COMMENT:**

*2- The authors show a lot of data from flow cytometry experiments, however the plots and gate*
*strategy are not shown anywhere. It would be advisable to show that, for instance, for data in*
*Figure 2, figure 5f, figure 6c and 6d, figure 7c*

**AUTHOR RESPONSE:**

We thank the reviewer for this comment and have added all recommended gating plots in
Figure 5c, 5f, and Figure SI 2a, 2d, 2k, 2i, 5c, 6d, 6e, 7c in the revised manuscript.

**REVIEWER COMMENT:**

*3 – There is data in the literature showing that CXCL1 and CXCL2 Regulate NLRP3*
*Inflammasome activation via CXCR2 (J Immunol September 1, 2017, 199 (5) 1660-1671).*
*Curiously, it seems that EAE-ELT mice display reduced IL-1beta levels and that this EAE*
*phenotype is ASC independent. What would be the authors comment on that? In addition, do*
*myeloid cells also display increased CXCR2 expression in EAE-ELT mice?*

**AUTHOR RESPONSE:**

We thank the reviewer for bringing to our attention the J Immunology article (2017, 199 (5)
1660-1671). Indeed, we found that myeloid cells such as dendritic cells and macrophages
display increased CXCR2 expression in ELT-EAE mice (shown in Figure SI 5c in the revised
manuscript). Our result indicates that it is possible that the inflammasome independent
mechanism of ELT-EAE is driven by upregulated myeloid CXCR2 signaling. We added relevant
discussion points to comment on “why ELT-EAE mice have reduced IL-1 β and ASC
dependency” in the revised manuscript, lines 381-386.

**Reviewer #2, expert on trauma and stress responses (Remarks to the Author):**

*Khaw and colleagues studied the influence of early life trauma (ELT) on experimental immune*
*encephalitis (EAE) phenotype in mice. The study suggests immune cell β 1 adrenergic signaling*
*as a druggable molecular target to interfere with ELT-EAE phenotype.*
*Not surprisingly, there is only limited knowledge about the relation between ELT and MS in*
*humans, since approaches to study this relation would require a sound understanding in how far*
*MS patients suffered from ELT – such an approach appears virtually impossible to conduct, not*
*only because of practical and ethical concerns. The study at hand will broaden the view*
*concerning the heterogeneity of MS and its relationship with an additional environmental factor*
*(ELT). Clearly, this is of importance, as it may help to increase the quality of MS treatment and*
*to assess disease progression. The work might give interesting implications for an option*
*treating MS patients, who do not respond to IFN β therapy. Furthermore, the paper suggests*
*biomarkers that appear promising for MS studies in humans.*
*The manuscript is well written, and the authors have undertaken a substantial amount of work to*
*provide evidence for their findings. The presentation of the results is clear and comprehensible.*
*The authors discussed their findings in the context of the existing and relevant literature.*
*The authors may wish to consider the following points to further improve their manuscript:*

We thank the reviewer 2 for his/her positive comments. According to reviewer 2's suggestion,
we conducted experiments. Details are shown in below. It is our belief that the manuscript is
substantially improved after making the suggested edits.

**REVIEWER COMMENT:**

*The authors used a model to create ELT by 'subjecting mouse pups to maternal separation with*
*phosphate buffered saline injections'. Please explain the usage of PBS injection additional to*
*maternal separation. I also did not understand the rational of the sentence 'First, mouse pups*
*were separated daily from their mother and father.' Do the authors normally breed newborn*
*mice together with the father?*

**AUTHOR RESPONSE:**

We thank the reviewer for allowing us to improve the quality of our model description. Because
emotional neglect and physical abuse affect MS disease¹, we selected an ELT paradigm that
involves aspects of both psychological and physical abuse through maternal separation and
daily injections, respectively². From additional studies (please see response to next comment)
conducted with further control mice, namely, neonates were subjected to either maternal
separation or PBS injection alone, we observed that the entire disease phenotype of ELT-EAE
cannot be recaptured if Msep or PBS injection alone was applied prior to EAE induction. Thus,
we concluded that both Msep and PBS injection stressors (ELT) are necessary for the
development of the unique phenotype. On a separate note, newborn mice share their cage with
parents until wean time. We added these issues to the revised manuscript.

**REVIEWER COMMENT:**

*I am wondering that only one control group was used for the data presented in Figures 1-3.*

*There should be three control groups, since it remains questionable, whether the measured*
*effect (immune phenotype) results from the combination of maternal deprivation and injection*
*(as stated by the authors), or whether it is simply the result of the injection alone. The authors*
*should use the following three control groups: i) no handling, ii) maternal deprivation, iii) ip.*
*injection alone.*

**AUTHOR RESPONSE:**

We totally agree with the reviewer's point about control groups. By virtue of the reviewer's
comment, we prepared "Msep alone EAE" and "PBS alone EAE" as further control groups, and
conducted additional experiments on EAE severity and duration in both male and female
(controls for Figure 1 study), demyelination analysis (controls for Figure 1 study), lymph node-
derived immune cell populations (controls for Figure 2 study), myeloid co-stimulatory molecules
expression (controls for Figure 2 study), CNS neuron status (controls for Figure 3 study) as well
as IFN β sensitivity (controls for Figure 4 study). We found "PBS alone" groups showed similar
phenotype of "no handling" control group. On the other hand, "Msep alone" group showed
similar phenotypes of ELT (Msep+PBS) group in EAE disease duration, CNS neuron damage,
and IFN β sensitivity. However, "Msep alone" group do not show similar phenotypes of ELT
group in demyelination status, lymph node-derived immune cell populations, and myeloid co-
stimulatory molecules expression. Our result suggests that ELT-EAE phenotype may be
exclusively presented by the specific conditions of subjecting B6 animals to neonatal Msep and
PBS injection followed by EAE induction in adulthood. We added these findings to the revised
manuscript (Fig. SI 1a, b, c; Fig. SI 2c, g; Fig. SI 3a, and Fig. SI4).

**REVIEWER COMMENT:**

*Since many factors are influencing the variability in ELS animal models (summarized in*
*PMC6079200) the method section should be further specified. Detailed information according*
*ARRIVE criteria are missing (e.g. clear background information of the used mouse strains:*
*C57B6N or C57B6J, number of animals, litter size, sex ratio per litter). Was the approach*
*"control vs ELT treatment" conducted in the same litter or different litters? Were the animals in*
*Fig 5-8 male or female mice?*

**AUTHOR RESPONSE:**

We thank the reviewer for this comment. We added all relevant information to methods section
under the subtitle "Mouse" and figure legends.

**REVIEWER COMMENT:**

*Additional methodological information should be supplied, e.g. for the PCR method. Which*
*primer sequences and housekeeping genes have been used? Which vehicle substance was*
*used for the pharmaceutical treatment of AR agonist and antagonist?*

**AUTHOR RESPONSE:**

We thank the reviewer for this comment. We added all relevant information about reagents,
antibody, ELISA kit, and primer sequence in Table S1 and S2. We also added information of

vehicle for all reagent to the method section of the revised manuscript.

**REVIEWER COMMENT:**

*Statistics: The authors assume a normal distribution of the data though they did not test for*
*normal distribution. Since the authors present SEM instead of SD, the exact n for each*
*experimental group must be indicated in the figure legends. In addition, the text in the section*
*“statistical analysis” is identical with the text in another paper – Please reword to avoid self-*
*plagiarism.*

**AUTHOR RESPONSE:**

We are grateful for the reviewer’s crucial comments which greatly improved our manuscript
upon addressing them in our revised manuscript. We have provided the exact “n” for each
experimental group. We have also rewritten the mentioned methods section.

**REVIEWER COMMENT:**

*Figure 3g: IBA1 stains both resident microglia and blood derived macrophages, which invade*
*the brain during EAE. Both populations differ in phenotype and function. As shown in Figure 2f,*
*the authors distinguished between macrophages and microglia in the EAE affected spinal cord.*
*Microglia / macrophages express adrenoceptors and could respond to altered NE levels due to*
*SNS hyperactivity. Is there anything known about the distinct roles and activation pattern of this*
*two populations in the EAE affected tissue? Could microglia / macrophages be the link between*
*ELT-EAE and increased severity and susceptibility in EAE? Please comment.*

**AUTHOR RESPONSE:**

We are grateful for the reviewer’s comment. It is known that ELT interferes with microglial
developmental programs, including their proliferation and death and their phagocytic activity^{3, 4}.
Recent studies suggest that ELT can cause changes in microglia transcriptomics that persist
into adulthood⁵. These changes are linked to altered microglial behavior in homeostasis and
inflammation contexts. Thus, as the reviewer pointed out, it may be that adrenergic receptor
expressing microglia/macrophages may be instrumental in the development of severe EAE
symptoms in adulthood. We discussed relevant recent literature to better inform the
implications of our current work in revised discussion (line 454-464).

**REVIEWER COMMENT:**

*Figure 6 shows plasma and lymph node NE levels in adult control and ELT mice. Is there any*
*information about peripheral and central NE levels in control and ELT mice after EAE induction?*
*I wonder why the authors stopped measuring before EAE induction? Please comment.*

**AUTHOR RESPONSE:**

We thank the reviewer for raising this question. We planned to measure NE levels before EAE
initially to answer the question of “does ELT affect systemic NE levels which is reflective of SNS
dysregulation?”. By virtue of this reviewer comment, we conducted the additional experiment of
evaluating NE levels in ELT-EAE mice and found that NE is also significantly greater in ELT-

EAE mice when compared to control EAE mice. New data are shown as Figure SI 6b in the
revised manuscript.

**REVIEWER COMMENT:**

*Figure 4b: Do female control EAE mice show the same EAE score pattern than male control*
*EAE mice with and without IFN β treatment?*

**AUTHOR RESPONSE:**

We thank the reviewer for this important question. We conducted this experiment and found
that female control EAE mice show similar EAE score pattern than male control EAE mice with
and without IFN β treatment. We added data as Figure 4b in revised manuscript.

**REVIEWER COMMENT:**

*Suppl Fig 5: Serum corticosterone levels did not differ between control and ELT group in EAE*
*mice. Since corticosterone release follows a circadian rhythm, the time of sampling influences*
*the results. Please state whether the samples were taken at the beginning or the end of the*
*activity phase of the mice.*

**AUTHOR RESPONSE:**

We thank the reviewer for this comment. Blood samples were always taken at approximately 9
am to standardize environmental condition. Our animals are housed in a pathogen free facility
with a 12 h light/dark (6am/6pm) cycle, thus at 9 am (3 hours after end of active period), animals
are expected to have lower baseline corticosterone levels. Statement is added to the method
section of the revised manuscript.

**REVIEWER COMMENT:**

*The rationale for using ASC $^{-/-}$ mice is unclear. Please add an appropriate introduction of this*
*approach and explain the concept of inflammasome dependent and independent EAE in more*
*detail.*

**AUTHOR RESPONSE:**

We agree with the reviewer that an appropriate introduction to the approach was lacking. We
previously published that control EAE disease was abolished in inflammasome-deficient Asc $^{-/-}$
mice and by IFN β treatment⁶, and that IFN β suppresses inflammasome activity⁷. Because we
found ELT-EAE is an IFN β -resistant, we speculated that ELT-EAE mice is induced by
inflammasome-independent signal. To confirm this speculation, we conducted experiment using
Asc $^{-/-}$ mice. We have added additional background to revised text (line 194-198).

**REVIEWER COMMENT:**

*Abstract: "...however, the underlying mechanism is yet well defined." Probably the authors want*
*to express "...the underlying mechanism is not yet well defined. "?*

**AUTHOR RESPONSE:**

We thank the reviewer for providing this kind and helpful suggestion. We have adjusted text in
the revised document to reflect the suggested change.

**References:**

1. Spitzer, C. *et al.* Childhood trauma in multiple sclerosis: a case-control study. *Psychosom Med*
**74**, 312-318 (2012).

2. Burenkova, O.V., Aleksandrova, E.A. & Zarayskaya, I.Y. Effects of early-life stress and HDAC
inhibition on maternal behavior in mice. *Behav Neurosci* **133**, 39-49 (2019).

3. Chocyk, A. *et al.* Maternal separation affects the number, proliferation and apoptosis of glia cells
in the substantia nigra and ventral tegmental area of juvenile rats. *Neuroscience* **173**, 1-18
(2011).

4. Reus, G.Z. *et al.* Early Maternal Deprivation Induces Microglial Activation, Alters Glial Fibrillary
Acidic Protein Immunoreactivity and Indoleamine 2,3-Dioxygenase during the Development of
Offspring Rats. *Mol Neurobiol* **56**, 1096-1108 (2019).

5. Delpech, J.C. *et al.* Early life stress perturbs the maturation of microglia in the developing
hippocampus. *Brain Behav Immun* **57**, 79-93 (2016).

6. Inoue, M., Williams, K.L., Gunn, M.D. & Shinohara, M.L. NLRP3 inflammasome induces
chemotactic immune cell migration to the CNS in experimental autoimmune encephalomyelitis.
*Proc Natl Acad Sci U S A* **109**, 10480-10485 (2012).

7. Inoue, M. *et al.* Interferon-beta therapy against EAE is effective only when development of the
disease depends on the NLRP3 inflammasome. *Sci Signal* **5**, ra38 (2012).

REVIEWERS' COMMENTS

Reviewer #1 (Remarks to the Author):

The revised version of the manuscript is substantially improved and addressed all the issues raised before. No additional concerns.

Reviewer #2 (Remarks to the Author):

Khaw and colleagues intensively revised their manuscript and brought further relevant information and evidence.

Given the fact that a lot of different models to induce early life trauma and adult stress exist, the model used here combines physical and psychological stressors.

In this work, the authors present further mechanistical evidence how stress can influence physiological functions, here, specifically with a focus on the immune system.

This work is important to the field of neuro-psychiatric research and to the MS field because it shows a further example how stress and disease development are linked together.